# CogVideo: Large-scale Pretraining for Text-to-Video Generation via Transformers

**Wenyi Hong**[†][*] **Ming Ding**[†][*] **Wendi Zheng**[†] **Xinghan Liu**[†] **Jie Tang**[†‡]
[†]Tsinghua University  [‡]BAAI
{hwy22@mails, dm18@mails, jietang@}.tsinghua.edu.cn

## Abstract

Large-scale pretrained transformers have reached a milestone in text (GPT-3) and text-to-image (DALL-E and CogView) generation. However, its application to video generation still has several challenges: *unaffordable* huge computation cost and *scarcity and weak* relevance of the text-video datasets. In this work, we present CogVideo, a 9B-parameter transformer for text-to-video generation. The CogVideo model has been trained by inheriting a pretrained text-to-image model, CogView2, which significantly reduces the training cost and alleviates the problem of scarcity and weak relevance. We also propose a multi-frame-rate training strategy for better aligning text and video clips. CogVideo achieves state-of-the-art performance in machine evaluation and outperforms publicly available models by a large margin in human evaluation. Its codes and model are also publicly available at `https://github.com/THUDM/CogVideo`.

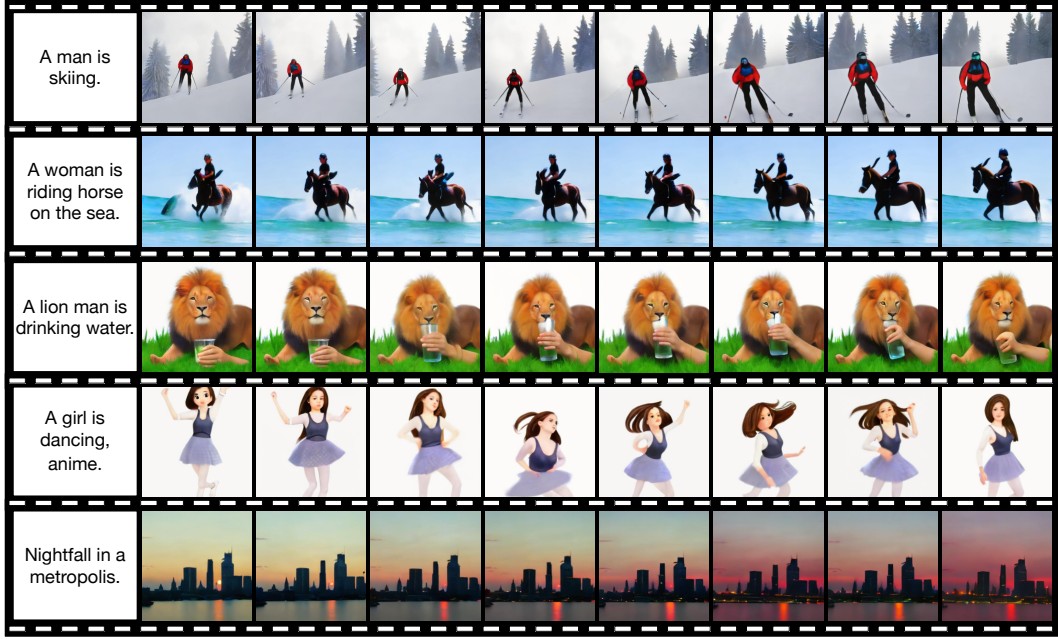

Figure 1: Samples generated by CogVideo. The actual text inputs are in Chinese. Each sample is a 4-second clip of 32 frames, and here we sample 8 frames uniformly for display purposes.

## 1 Introduction

Autoregressive transformers, e.g. DALL-E (Ramesh et al., 2021) and CogView (Ding et al., 2021), have revolutionized text-to-image generation. A few other works have also followed the framework to develop text-to-video transformers (Wu et al., 2021b; Ge et al., 2022), e.g. VideoGPT (Yan et al., 2021), and demonstrated its superiority over GAN-based methods (Clark et al., 2019; Tulyakov et al.,

---
[*]Equal contribution.

2018). However, the performances are still far from satisfactory. Diffusion probabilistic models, e.g. Imagen (Saharia et al., 2022) and DALLE-2 (Ramesh et al., 2022), represent another line of research for text-to-image generation and video generation Ho et al. (2022). However, how to better incorporate the temporal information for text-to-video generation is still a challenge.

In this paper, we focus on designing an autoregressive model for text-to-video generation. The critical challenge in previous work is that the generated video frames tend to gradually deviate from the text prompt. This makes vanilla autoregressive models only good at synthesizing videos with regular (e.g. forward moving cars) or random patterns (e.g. speaking by random moving lips), but fail at text prompt such as "a lion is drinking water". The main reason is that in the former case the first frame already provides sufficient information for the subsequent changes, while in the latter the model has to precisely understand the action "drink" in order to correctly generate the desired action — the lion lifts the glass to its lip, drinks and then puts down the glass.

Why could the autoregressive transformers well understand the text-image alignment, but struggle for the text-action alignment in videos? One fact is that the duration of videos varies a lot. Previous models split the video into many clips with a fixed number of frames for training (Wu et al., 2021b; Ge et al., 2022). Such treatment destroys the alignment between the text and its temporal counterparts in the video. If a "drinking" video is split into four individual clips of "holding a glass", "lifting", "drinking" and "putting down" with the same text "drinking", the model will be confused to learn the precise meaning of drinking.

The other challenge is that the perfect aligned text-video data is scarce, compared to the easy-to-collect billions of text-image pairs (Ramesh et al., 2021). VATEX is probably the largest annotated text-video dataset (Wang et al., 2019). However, it has only 41,250 videos. The retrieval-based text-video pairs, e.g. Howto100M (Miech et al., 2019), are weakly relevant and most captions only describe the scene without temporal information.

**Present Work.** Here we present a large-scale pretrained text-to-video generative model, CogVideo, which is of 9.4 billion parameters and trained on 5.4 million text-video pairs. To reduce the computational cost, CogVideo has been developed to inherit the knowledge learned from a text-image pretraining model CogView2 (Ding et al., 2022). To ensure the alignment between text and its temporal counterparts in the video, we propose the *multi-frame-rate training*. The flexibility of the textual condition makes it possible to simply prepend a piece of text describing the frame rate to the original text prompt for modeling different frame rates. To keep the text-video alignment, we choose a proper frame rate description to include the complete action in each training sample. The frame rate token also controls the intensity of the changes throughout continuous frames in generation. We train a sequential generation model and a frame interpolation model. The former model generates key frames according to the text, and the latter recursively fills the middle frames by varying the frame rates to make the video coherent. As shown in Figure 1, CogVideo can generate high-resolution (480×480) videos. The human evaluation demonstrates that CogVideo outperforms most publicly available models by a large margin. Our main contributions include:

- We present CogVideo, which is the **largest** and **open-source** pretrained transformer for general text-to-video generation. CogVideo demonstrates state-of-the-art FVD on the UCF-101 benchmark.

- We propose the multi-frame-rate training to better align text-clip pairs, which **significantly improves the generation accuracy**, in particular for movements of complex semantics. This training strategy offers CogVideo the capacity of controlling the intensity of changes during the generation.

- We design dual-channel attention to elegantly and efficiently finetune a pretrained text-to-image generative model for text-to-video generation, **avoiding the expensive full** parameter pretraining from scratch.

## 2 RELATED WORK

### 2.1 VIDEO GENERATION

Video generation is a long-standing research topic. Most previous works focus on the next-frame prediction task — forecasting the future frames based on the first video frame. Early works, e.g.

CDNA (Finn et al., 2016) and PredRNN (Wang et al., 2017), leverage deterministic methods to directly predict the next frame via CNNs or RNNs. However, these deterministic models are unable to capture the stochastic temporal patterns and synthesize coherent complex scenes. Recently, generative models, especially Generative Adversarial Networks (Goodfellow et al., 2014) (GANs), begin to dominate the area as they can perform unconditional or class-conditional video synthesis without the first frames. VGAN (Vondrick et al., 2016) is the first one to use GAN for video generation. It decomposes video to a static background and a moving foreground, and then generates them with 2D and 3D convolutional networks respectively. TGAN(Saito et al., 2017) proposes to separately generate the temporal latent variables and spatial information, and MoCoGAN (Tulyakov et al., 2018) similarly decomposes the latent space into context and motion subspaces. DIGAN (Yu et al., 2022) applies implicit neural representations for video encoding. Recently, text-to-video generation emerges as a promising direction. The framework of VQVAE (van den Oord et al., 2017) and autoregressive transformers (Vaswani et al., 2017; Brown et al., 2020) quickly become the mainstream methods (Wu et al., 2021a;b; Ge et al., 2022). Ho et al. (2022) proposes a video diffusion model along with a gradient method recently for text-to-video generation. The previous methods are basically trained on a specific dataset, e.g. UCF-101 (Soomro et al., 2012), making the trained model domain-specific. Moreover, most of these models are not publicly available.

## 2.2 AUTOREGRESSIVE TRANSFORMER

Recent years have witnessed the autoregressive transformer emerging as a powerful generative model. The autoregressive models become the most prevalent framework for text generation (Sutskever et al., 2011). With its prominent capacity of fitting, transformer (Vaswani et al., 2017) gradually becomes a standard neural structure for text generation. Examples are GPT-3 (Brown et al., 2020) and GLM-130B (Zeng et al., 2023). In computer vision, van den Oord et al. (2017) first proposes to train a VQVAE to compress the image into a sequence of tokens from a learned dictionary, which can be then efficiently handled by the autoregressive model. VQ-GAN (Esser et al., 2020) learns a more semantic-aware dictionary for unconditional image generation. In the text-to-image generation, pretrained autoregressive transformers such as DALL-E (Ramesh et al., 2021) and CogView (Ding et al., 2021) have shown superiority in open-domain image generation. Besides the pure GPT-style generation, CogView2 (Ding et al., 2022) proposes a new language model CogLM for infilling in the image generation. Recent autoregressive transformers (Rakhimov et al., 2020; Yan et al., 2021; Wu et al., 2021a;b) have also shown their superiority in video generation. Among them, GODIVA (Wu et al., 2021a) and NÜWA (Wu et al., 2021b) focus on the open-domain text-to-video generation. However, they simply generate frames or frame blocks one by one in chronological order, and suffer from poor text-video alignment (Cf. § 1).

Diffusion probabilistic models, e.g. Imagen (Saharia et al., 2022) and DALLE-2 (Ramesh et al., 2022), recently showed very promising results in text-to-image generation. However, in this paper we mainly focus on autoregressive models due to the sequential nature of temporal information.

## 3 METHOD

We first introduce *multi-frame-rate training* to better align text and video semantics (§ 3.1). To overcome the data scarcity and accelerate pretraining, we propose efficient *dual-channel attention* for video generation by inheriting knowledge from a pretrained text-image model (§ 3.2).

## 3.1 MULTI-FRAME-RATE TRAINING

Before the training, we first tokenize each frame into image tokens, a similar strategy also used in the framework of VQVAE (van den Oord et al., 2017).

**Training.** The key design here is that we add a variable frame-rate token to the text and sample frames at this frame rate to compose a fixed-length training sequence. The motivations are two folds:

1. Directly separating the long video into clips at a fixed frame rate often leads to semantic mismatching between clips and captions. The truncated clip might only contain incomplete actions, which do not correspond to the full text.

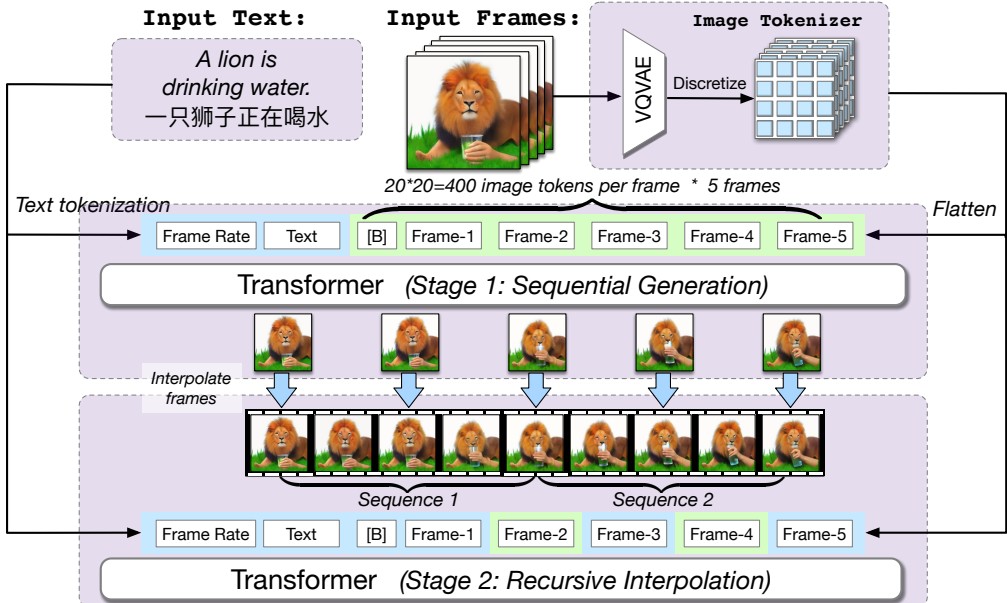

Figure 2: Multi-frame-rate generation framework in CogVideo. Input sequence includes frame rate, text, and frame tokens. [B] (Begin-of-image) is a separator token, inherited from CogView2. In stage 1, $T_s$ frames are generated sequentially on the condition of frame rate and text. Then in stage 2, generated frames are re-input as bidirectional attention regions to recursively interpolate frames. The frame rate can be adjusted during both stages. Bidirectional attention regions are highlighted in blue , and unidirectional regions are highlighted in green .

2. The adjacent frames are usually very similar. A sudden change from the previous frame may incur a large loss. This will lead the models less inclined to explore the long-range correlation because simply copying the previous frame acts like a shortcut.

Therefore, for each training sample, we align the text and the frames by sampling videos at variable frame rates, so that videos of any length are evenly down-sampled to $T_s$ frames. In practice, we set up a series of predefined frame rates, and select the lowest frame rate for each text-video pair at which we can sample at least $T_s$ frames. For each video, the frame rate information is described in the form of text and prepended to the original text.

Although the above method improves the alignment of text and video, a side effect is that the generated video at a low frame rate could be incoherent due to the large changes between frames. We thus train another *frame interpolation* model to insert transition frames to the generated samples of the sequential generation model. Frame interpolation relies heavily on bidirectional information. However, previous transformers (Wu et al., 2021a; Yan et al., 2021; Wu et al., 2021b) are mostly unidirectional. To be aware of the bidirectional context, we adopt Cross-Modal General Language Model (CogLM) Ding et al. (2022) which unites bidirectional context-aware mask prediction and autoregressive generation by dividing tokens into unidirectional and bidirectional attention regions. A token in a unidirectional region can attend to the tokens in all bidirectional regions and previous unidirectional regions, while a token in bidirectional regions can only attend to the tokens in all bidirectional regions. As shown in Figure 2, (1) *all frames* in stage 1 and *the 2nd, 4th frames* in stage 2 are in the unidirectional region; (2) {Frame Rate}, {Text} and all *other frames* belong to the bidirectional region.

In this way, bidirectional attention context in text and given frames is fully exploited without interfering with auto-regressive frame prediction. The models of the two stages can also share the same structure and the training process only with different attention masks.

**Generation.** The multi-frame-rate generation is a hierarchical and recursive process, illustrated in Figure 2. Specifically, the generation pipeline consists of a sequential generation stage and a recursive interpolation stage:

1. Sequentially generates $T_s$ key frames based on a low frame rate and text. The input sequence is `[{Frame Rate}{Text} [B] {Frame1} ... {Frame $T_s$}]`. In practice, we set $T_s = 5$ and the minimum sampling frame rate to 1 frame per second (fps). The hyperparameters are determined by the memory of devices and the performance in small primary experiments.

2. Recursively interpolate frames based on the text, frame rate and known frames. In each round of interpolation, we split the generated frames into multiple $\lceil \frac{T_s}{2} \rceil$-frame blocks overlapping at the beginning and the end, and interpolate a frame between the successive frames in each block. The input sequence to CogLM is also `[{Frame Rate}{Text} [B] {Frame1} ... {Frame $T_s$}]`, where Frame $2i (i = 1, 2, ..., \lfloor \frac{T_s}{2} \rfloor)$ are to be autoregressively generated. By recursively doubling `{Frame Rate}`, we can conduct finer interpolation to generate videos of many frames.

## 3.2 DUAL-CHANNEL ATTENTION

Large-scale pretraining usually demands a large dataset. For the open-domain text-to-video generation, ideally we hope the dataset contains sufficient text-video pairs so as to infer both spatial and temporal correlation between video and text. However, it is rather expensive and time-consuming to collect high-quality text-video pairs. A compromise method is to leverage the image data to facilitate the learning of spatial semantics. Video Diffusion Model (Ho et al., 2022) and NÜWA (Wu et al., 2021b) try to add text-image pairs into text-video training, which helps achieve better results. However, adding image data will also significantly increase the training cost, especially in large-scale pretraining scenarios.

In this paper, we propose to inherit the learned knowledge from a pretrained text-to-image generation model rather than use the raw image data. Pretrained text-to-image models, e.g. CogView2 (Ding et al., 2022), already have a good command of the text-image relations. The coverage of the data used to train these models is also larger than that of videos.

How to train a video generation model on top of an image generation model? The proposed technique is *dual-channel attention*. As shown in Figure 3, we augment the original attention block (the spatial channel) with an additional cross-frame attention block (the temporal channel). The temporal channel is implemented by a 3D Swin attention (Liu et al., 2021). Besides the restriction of the receptive window, the temporal channel also follows the attention mask of CogLM. More specifically, during sequential generation, a token attends to the tokens in the previous frames and tokens before it in this frame; and during frame interpolation, the tokens additionally attend to the known frames.

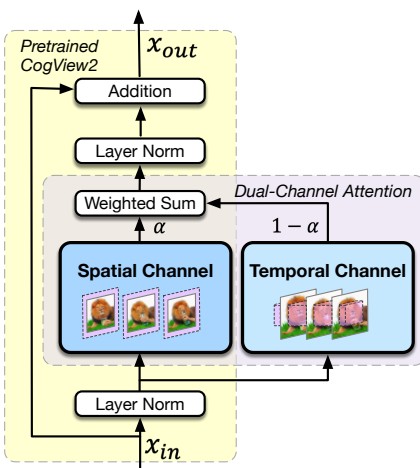

We freeze all the parameters in the pretrained text-to-image model to preserve its learned knowledge, and finetune the parameters in the temporal channel attention block to learn the cross-frame information. Specifically, the dual-channel attention block with Sandwich-LN (Ding et al., 2021) can be formulated as

Figure 3: The dual-channel attention. The parameters in CogView2 are frozen.

$$x = \boldsymbol{\alpha} \cdot \text{SpatialAttn}(\text{LayerNorm}(\boldsymbol{x_{in}})) + (\boldsymbol{1} - \boldsymbol{\alpha}) \cdot \text{TemporalAttn}(\text{LayerNorm}(\boldsymbol{x_{in}})), \quad (1)$$

$$\boldsymbol{x_{out}} = \boldsymbol{x_{in}} + \text{LayerNorm}(\boldsymbol{x}). \quad (2)$$

The inter-channel mixture factor $\boldsymbol{\alpha}$ is a vector $\in (0, 1)^d$, where $d$ is the hidden size of the input feature $\boldsymbol{x_{in}}$. To restrict the range of $\boldsymbol{\alpha}$ within $(0, 1)^d$, we reparameterize it as $\boldsymbol{\alpha} = \text{sigmoid}(\boldsymbol{a})$, where $\boldsymbol{a} \in \mathbb{R}^d$ is a learnable parameter. Thanks to the decomposition of attention channels in *dual-channel attention*, the computation procedure and hyper-parameters of each channel can be flexibly and independently adjusted under various tasks, as long as the outputs of both channels share the same hidden size. We initialize the temporal channel the same as the pretrained spatial channel so that the

attention output $x$ is still in the original CogView2 feature space at the beginning of finetuning. A detailed analysis of the attention is in Appendix B.

As mentioned above, we adapt Swin Attention (Liu et al., 2021) for the temporal channel attention to reduce the time and memory overhead. An interesting finding is that, the Swin attention provides a chance for parallel generation in faraway regions of different frames, which further accelerates the autoregressive generation. Further details and acceleration results are illustrated in Appendix A.

## 4  PRETRAINING

**Model.**  CogVideo consists of two models corresponding to two stages. The backbone of the model in each stage is a 48-layer Transformer with dual-channel attention, with 48 attention heads in each attention channel and a hidden size of 3,072. Each of the two models has 7.7 billion parameters, while 6 billion of them are shared between both models, which finally results in a total of 9.4 billion parameters in CogVideo. To avoid potential gradient explosion, we also use Sandwich LayerNorm and use PB-Relax for stabilizing training, as suggested in (Ding et al., 2021). Shifted window attention is adopted in the Stage 2 model with a window size of $10 \times 10$.

**Dataset.**  We pretrain our model on a dataset of 5.4 million text-video pairs with a resolution of $160 \times 160$ (can be upsampled to $480 \times 480$ further). The data is mainly crawled from the Internet, where each video has its matching caption. About 30% of the captions are in English, which have been translated into Chinese by machine translation. About 50% of the captions are sentences, while the others are made of phrases.

**Pretraining.**  Sequences in both stages are of the length 2,065, consisting of 64 text tokens, 5 (frames) $\times$ 400 (per frame) image tokens, and 1 separator token. Both text and images are tokenized using icetk[1]. The model in stage 1 is first pretrained for 76,000 iterations on video clips with a minimum frame rate of 0.25 fps, then trained for 15,000 iterations with a minimum frame rate of 1 fps. The model in stage 2 is pretrained for 78,500 iterations with frame rates of 2, 4, and 8 fps. Both models are trained in FP16 with batch size $= 416$, and optimized by Adam with max learning rate $= 2 \times 10^{-4}$, $\beta_1 = 0.9$, $\beta_2 = 0.95$, weight decay $= 1 \times 10^{-2}$.

## 5  EXPERIMENTS

### 5.1  MACHINE EVALUATION

Machine evaluation was conducted on two popular benchmarks for video generation, UCF101 (Soomro et al., 2012) and Kinetics-600 (Carreira et al., 2018). Following Rakhimov et al. (2020); Yu et al. (2022), we use Fréchet Video Distance (FVD) (Unterthiner et al., 2018) and Inception Score (IS) (Salimans et al., 2016) as evaluation metrics. IS is calculated based on the C3D model (Tran et al., 2015) which was first trained on the Sports-1M dataset (Karpathy et al., 2014) and then finetuned on the UCF101 dataset[2]. FVD is calculated based on I3D model (Carreira & Zisserman, 2017) trained on Kinetics-400, following previous works (Yu et al., 2022; Ge et al., 2022)[3]. As the low-level features brought by the image tokenizer (VQ-VAE) may increase the distribution difference between generated samples and ground truth, which is not the focus of this work, we also evaluate FVD using ground truth reconstructed by the tokenizer.

**UCF-101** is a human action dataset consisting of 13,320 videos annotated with 101 action classes. Due to the gaps in image style and frame rate between CogVideo's training set and UCF-101, we finetune CogVideo for 10,000 iterations with a batch size of 192. The model is trained on the whole dataset, following the setting of Clark et al. (2019), Yan et al. (2021), Tian et al. (2021), Yu et al. (2022). We use class labels as input text and generate samples according to the class distribution during inference. For a fair comparison with previous works, we follow Ge et al. (2022) to resize the original $160 \times 160$ CogVideo generation to $128 \times 128$, and evaluate FVD and IS over 2,048 and

---

[1]`https://github.com/THUDM/icetk`

[2]We evaluate IS and FVD with the official code of TGAN-v2 and TATS respectively.

[3]In a previous version of this paper, we had a bug in the FVD evaluation of both UCF-101 and Kinetics-600 due to using a wrong I3D checkpoint. We corrected the results in this version.

Table 1: Video generation performance on UCF-101. Class labels are used as the text inputs. * means trained on the training split of UCF-101 only. ** means that the ground truth used in FVD is reconstructed by the tokenizer.

| Method | IS (↑) | FVD (↓) |
|---|---|---|
| VideoGPT (Yan et al., 2021) | 24.69 | - |
| DVD-GAN (Clark et al., 2019) | 27.38 | - |
| TGANv2 (Saito et al., 2020)* | 28.87 | 1209 |
| MoCoGAN-HD (Tian et al., 2021) | 32.36 | 838 |
| DIGAN (Yu et al., 2022)* | 29.71 | 655 |
| DIGAN (Yu et al., 2022) | 32.70 | 577 |
| TATS-base (Ge et al., 2022)* | 79.28 | 332 |
| CogVideo (Ours) | 51.11 | **305** |
| CogVideo (Ours)** | - | **220** |

Table 2: Video generation performance on Kinetics-600. The metrics are based on the 16-frame generated videos which prime on the first 5 frames, following settings of Rakhimov et al. (2020). ** means that the ground truth used is the reconstruction result of the tokenizer.

| Method | FVD(↓) |
|---|---|
| LVT (Rakhimov et al., 2020) | 224.73 |
| VT (Weissenborn et al., 2019) | 170 |
| DVD-GAN-FP (Clark et al., 2019) | 69.15 |
| TriVD-GAN-FP (Luc et al., 2020) | 25.74 |
| Transframer (Nash et al., 2022) | 25.4 |
| Video diffusion (Ho et al., 2022) | **16.2** |
| CogVideo (Ours) | 49.76 |
| CogVideo (Ours)** | **11.08** |

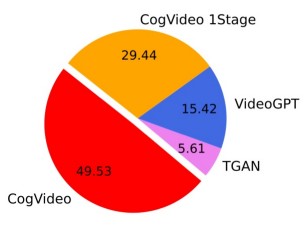
(a) Human preference. The percentage of being chosen as the best.

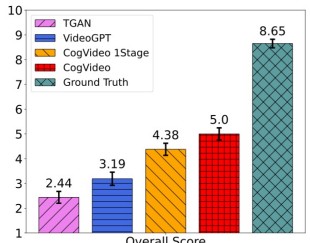
(b) Overall scores (1-10) for each method.

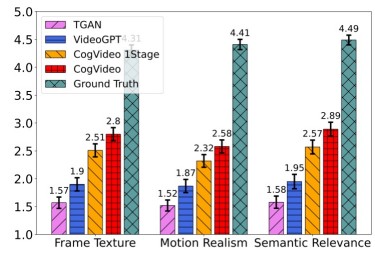
(c) Scores (1-5) on three important aspects.

Figure 4: Human evaluation results. "CogVideo 1Stage" refers to generating videos sequentially by circularly reinserting the last frame into CogVideo's Stage 1 model.

10,000 samples respectively. As shown in Table 1, our model achieves state-of-the-art FVD, and achieves higher IS than most baselines.

**Kinetics-600** contains 600 classes of human action videos, with roughly 350,000 train and 50,000 test videos in total. We use the action category as input text, and finetune CogVideo on the training set for 12,000 iterations with a batch size of 640. Following the setup of Weissenborn et al. (2019); Rakhimov et al. (2020), we center-crop and downsample each frame to 64×64 to measure FVD. Results are shown in Table 2. Our result underperforms some methods on the FVD of this dataset. However, we find that our method achieves much better FVD with reconstructed ground truth, which verifies the guess that the FVD performance is greatly influenced by the low-level features from the VQVAE tokenizer, although this influence is hard to distinguish by eyes after resized to $64 \times 64$ according to the setting (Weissenborn et al., 2019). Unfortunately, none of those previous models or their evaluation codes on Kinetics is open-source, which prevents us from further analyzing the reasons.

## 5.2 HUMAN EVALUATION

To further evaluate CogVideo, we invite 90 anonymous evaluators to rate for CogVideo and other open-source baselines including the GAN-based model TGANv2 (Saito et al., 2020) and the GPT-based model VideoGPT (Yan et al., 2021). 30 classes in UCF101 are randomly picked as text conditions and 4 aspects are rated. For VideoGPT, we use the official unconditional pretrained model[4]. For TGANv2, we use the official source code to train an unconditional generation model under the same setting as that in Saito et al. (2020). To assign unconditionally generated samples into corresponding categories, we choose TSM (Lin et al., 2019) as the action recognition model for post-classification. We only keep the samples whose likelihood of a certain class is at least 80%. See Appendix C for further details.

---

[4]https://github.com/wilson1yan/VideoGPT. We use the unconditional version of VideoGPT because the conditional checkpoint is not released.

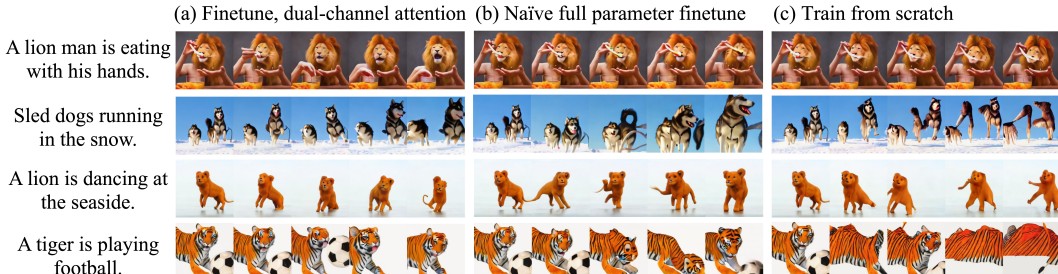

Figure 5: Ablation study for dual-channel attention. The samples are generated with a given first frame, 1 fps. Dual-channel attention produces more accurate shapes. The three models are trained on a "fast setting" with fewer data, iterations and batch size than CogVideo described in § 5.3.1.

Table 3: Ablation study for dual-channel attention on Kinetics-600. FVD is evaluated on generated 11-frame samples priming on 5 frames in a 5,000-sample subset of Kinetics-600's test set.

| Method | Initialization | Training Setting | FVD ($\downarrow$) |
|---|---|---|---|
| Train from scratch | Random | full parameter training | 166.13 |
| Finetune (Naïve) | CogView2 | full parameter training | 176.57 |
| Finetune (Dual-Channel) | CogView2 | only 1 channel trained | 124.92 |
| Finetune (Dual-Channel) | CogVideo | only 1 channel trained | 108.27 |

Results in Figure 4 show that CogVideo significantly outperforms baselines on multiple important aspects including frame texture, motion realism, and semantic relevance, and achieves the top score by the overall quality. 49.53% of evaluators choose CogVideo as the best method, while only 15.42% and 5.6% favor VideoGPT and TGANv2, respectively.

## 5.3 ABLATION STUDY

In this section, we conduct further studies on our two main technical contributions: dual-channel attention and multi-frame-rate training.

### 5.3.1 THE EFFECTIVENESS OF DUAL-CHANNEL ATTENTION

To verify the effectiveness of dual-channel attention, we conduct ablation studies on its two key components: (1) initializing with text-to-image pretrained model CogView2; (2) training temporal attention channel while freezing all other parameters. The former provides an initialization point with rich text-image knowledge, while the latter enforces preserving and exploiting that knowledge. Three settings are compared against (a) finetuning with dual-channel attention (the same way as CogVideo — initialized with CogView2 and only train temporal channel); (b) naïve full parameter finetune (initialize with CogView2 and remove temporal module, and apply full parameter finetuning); (c) training from scratch (randomly initialize and train all parameters).

First, we evaluate qualitatively on general-domain datasets. The models are trained on a 1 million subset of the pretraining dataset for 20,000 steps with a batch size of 256, where videos related to animals are removed to demonstrate the generalization ability. Generated samples are shown in Figure 5. Dual-channel attention outperforms the other settings with more accurate contours and details, such as the limbs of the "dancing lion" and the appearance of "sled dogs", while training from scratch performs the worst. As there is no animal-related video in this train set, the results indicate that both finetuning with dual-channel attention and naïve full parameter finetuning can transfer the knowledge from the text-to-image model, while the former one can better preserve the knowledge and thereby enhance training efficiency and performance.

Second, we further test the aforementioned 3 settings on Kinetics-600 for quantitative evaluation. We additionally test finetuning CogVideo to verify the effect of video pretraining. Each model is trained on Kinetics-600 train set for 11,000 iterations with a batch size of 160. Results were shown in Table 3, from which we can see that finetuning CogVideo scores the best, and finetuning with dual-channel attention get better FVD than both naïve full parameter finetuning and training from scratch, which indicates the superiority of dual-channel attention.

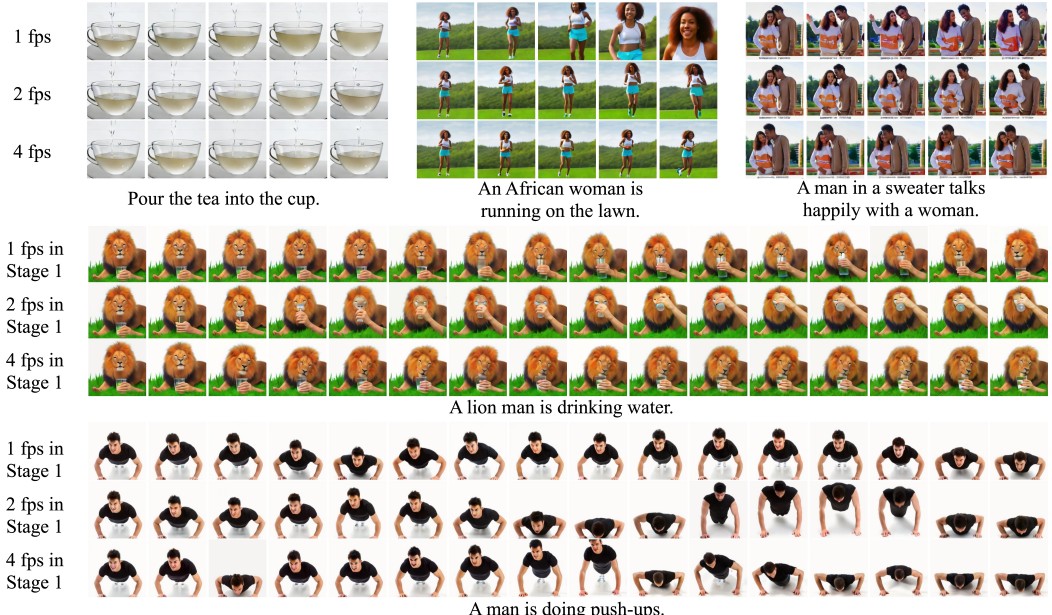

Figure 6: The *frame-rate token* controls the intensity of change during generation. (The top 3 groups) Generating the following 4 frames with different frame-rate tokens. (The bottom 2 groups) 4-second clips generated with different frame-rate tokens in Stage 1 and then interpolated to 16 frames.

### 5.3.2 CASE STUDIES OF MULTI-FRAME-RATE TRAINING

Multi-frame-rate training strategy not only enables CogVideo to generate videos of multiple frame rates, but also helps better align texts and videos of varying lengths. To demonstrate its effectiveness, we compare the samples generated with different frame rates in Stage 1 in Figure 6.

First, we use the Stage 1 model to generate 4 following frames conditioning on the 1, 2, and 4 fps tokens. A lower frame rate corresponds to a longer time interval between neighboring frames. For example, in Figure 6 (top 3 groups) the water level rises faster and the posture of people changes more between adjacent frames with a lower frame rate. These results verify that multi-frame-rate training is able to control the intensity of the changes throughout continuous frames with the frame rate tokens.

In order to verify how the multi-frame-rate training help align videos and texts, we process samples generated as above to the same duration and frame rate. To be concrete, we first extend samples to 4 seconds by circularly reinserting the last frame into the Stage 1 model, then interpolate to 4 fps with the Stage 2 model. As shown in Figure 6 (the bottom 2 groups), samples generated with a relatively low frame rate in the first stage may have more precise modeling of longer-term movements, e.g. the periodicity of push-ups and the whole process of "lifting a glass, drinking, then putting it down". On the other hand, CogVideo can generate realistic movements with a high frame rate in the first stage when only short-term dependency is needed, e.g. "talking". In other words, a higher frame rate is better at capturing the action details and modeling short-term actions, and a lower frame rate is more capable of capturing global information in the time dimension and modeling long-term actions.

## 6 CONCLUSION AND LIMITATIONS

We present CogVideo, to the best of our knowledge, the largest and the first open-source pretrained transformer for text-to-video generation in the general domain. CogVideo exhibits a way to efficiently leverage the pretrained text-to-image generative model for a text-to-video generation without hurting its image generation capacity. With the proposed multi-frame-rate training framework, CogVideo is endowed with a better understanding of text-video relations and abilities to control the intensity of changes during generation. There are still several limitations in CogVideo, e.g. restriction on the length of the input sequence still exists due to the large scale of the model and limitation of GPU memory, and we leave them for future work.

ACKNOWLEDGMENTS AND DISCLOSURE OF FUNDING

We would like to thank Zhao Xue, Shuai Zhao, Sha Yuan for their help in data collection, Weidong Guo, Fengyu Rao, Zhaoyang Zeng, Mingkang Tang, Zhuoyi Yang for their useful discussion, Hanxiao Qu for maintaining the machines and the computational resources supported by BAAI.

This research was supported by Natural Science Foundation of China for Distinguished Young Scholars 61825602.

## ETHICS STATEMENT

The primary goal of CogVideo is to advance research on video generation methods. Its ability of text-to-video generation has the potential for easing the effort of short video and digital art creation. While in the meantime, we are also aware of its possible ethical impact on society. it might be used for malicious purposes such as reinforcing social stereotypes, violating privacy, generating deceptive or harmful content, etc. In the following part, we discuss these issues and present possible solutions accordingly. Being aware of these ethical impacts, we set a license for CogVideo, which demands the users not to use CogVideo (or derivatives of the model) for any deeds that may violate laws or be harmful to the society.

**Problem 1: Reinforcing social stereotypes.** As mentioned in DALL-E2 (Ramesh et al., 2022) and Imagen (Saharia et al., 2022), visual generation models may inherit biases from their training data and reinforce social stereotypes. For example, if there are more male engineers than female engineers in the dataset, the pretrained model is inclined to generate males given input of "engineer". This problem can be alleviated to some extent by both pre-processing and post-processing. For pre-processing, we can fuzzy search keywords related to fairness (such as gender, race, and age) during data collection, and adjust their proportion. For post-processing, *although there are some researches on model post-processing for fairness, the methods for large pretrained model in the general domain is still an open problem.* A simple solution is Word Replacement proposed in Ding et al. (2021), based on the observation that the biases in the generated images/videos often come along with fuzzy user input. To be concrete, we can train an additional name entity recognition model to find the words about humans, then directly add accurate descriptions (such as "white", "black", "Asian" or "male", female") before those words. The descriptions are sampled according to the real proportion in the world.

**Problem 2: Violating privacy.** Researchers have found that some private information contained in the dataset could be extracted from pretrained language models (Carlini et al., 2021). The same problem exists in multi-modal pretrained models. As the datasets are mainly collected from the websites, private information may be included such as the user's image/video paired with their name. During data collection, we try to filter out data sources with such private information, though there inevitably remains a small number of videos of public figures.

**Problem 3: Generating deceptive or harmful content.** Although the generated videos still have a certain gap with real videos according to our human evaluation, we are conscious that pretrained models may generate realistic videos in the near future. Without adequate guardrails, such models could be used to intentionally misinform subjects and potentially empower information operations, or generate explicit content such as sexual and violent videos. During data collection, we manually filter out sources with inappropriate data including pornographic and violent content. We further filter out toxic texts using stop-word list and NSFW videos using models[5]. During the process we found that, in the original dataset, only 0.2% videos have NSFW value higher than 0.95 and only 0.5% captions contain stop words, while all of them are false positives according to manual inspection. During training, we choose to use CogView2 as our initialization, whose dataset doesn't contain sexual or violent images. When developing API, we set restrictions to prevent users from inputting harmful text descriptions. Additional classifiers will be trained along with pretraining to discriminate the fakes generated by a specific model.

Last but not least, CogVideo is committed to promoting academic research and will never be put into commercial use. And we choose not to release datasets in order to further ensure copyright issues.

---

[5]The model to filter out NSFW videos: `https://github.com/GantMan/nsfw_model`. We use the first frame to represent a video

## REPRODUCIBILITY STATEMENT

We have paid great exertion to ensure reproducibility.

- Open-source. We create an anonymous repository `https://anonymous.4open.science/r/CogVideo-anonymous-4148`, containing codes for pretraining and inference. As pretraining requires huge computational costs (pretraining CogVideo takes 20 days on 104 A100 GPUs), we also release checkpoints to the public to further ensure reproducibility.

- Details of models and training procedure: introduced in § 4. Also, all the details of the model (e.g. structures and settings) can be found in the released code.

- Dataset. A brief introduction can be found in § 4. Here we provide more details about the data and the way to reproduce CogVideo. Our data is crawled from public video websites, where each video is paired with a caption (either in English or Chinese). We filter out videos longer than 60 sec, as it may cause too weak relevance between the video and captions. The data covers multiple domains including natural scenery (42%), daily activity (36%), sports (5%), animals (7%), others (10%) (food, building, city, abstract art, etc.), and are mainly real videos (rather than artificial videos such as cartoon). The content and quality of crawled data are very similar to WebVid-10M[6], a recently released general domain text-video dataset, thus one can refer to WebVid-10M for reproducing.

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

APPENDIX

## A  SHIFTED WINDOW ATTENTION IN AUTO-REGRESSIVE GENERATION

To further alleviate the large time and memory overhead in the temporal channel during training and inference, we refer to Swin Attention Liu et al. (2021). The original Swin attention is only applied to

non-autoregressive scenarios, we extend it to the autoregressive and temporal scenarios by applying an auto-regressive attention mask in the shifted windows.

An interesting finding is that, **the Swin attention provides a chance for parallel generation in faraway regions of different frames**, which further accelerates the auto-regressive generation. The dependence of the generation of a specific token relies on

- Auto-regressive mask. A token can only attend to previous frames or tokens before itself in the current frame.
- Shifted window. Only tokens within the distance of window size in both width and height dimensions can be directly attended to.

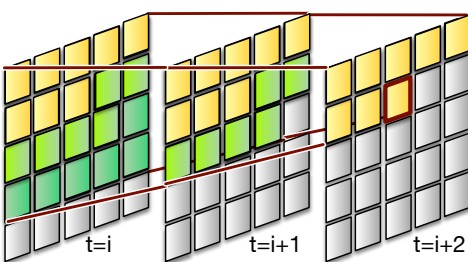

Figure 7: In 3D autoregressive swin attention (window size $2 \times 2$ as an example), the token in the red box can only attend to (either directly or indirectly) the yellow or green tokens. The gray tokens in the $i$-th frame and the token in the red box can be generated in parallel.

As shown in Figure 7, we can start generating parts of the tokens in the following frames before finishing the generation of all the previous frames — they can work in parallel. Suppose $X$,$Y$ is the height and width of each frame, and $A_x$, $A_y$ are the height and width of the shifted window. For two tokens at $(t_1, x_1, y_1)$ and $(t_2, x_2, y_2)$, $t_1 < t_2$, the latter cannot attend to the former either directly or indirectly if

$$(x_1 - x_2)Y + (y_1 - y_2) \geq (t_2 - t_1 + 1)(A_x Y + A_y), \tag{3}$$

which means that the $i$-th token in the $t$-th frame can be generated with the $(i - A_x Y - A_y)$-th token in the $(t+1)$-th frame in parallel. In this way, we can generate $\lfloor \frac{XY}{A_x Y + A_y} \rfloor$ tokens in parallel at most, thus greatly enhancing parallelism and accelerating inference compared to auto-regressive with standard attention which can only generate one token at a time.

To verify the acceleration effect provided by parallel generation, we generate 6 frames (32×32 tokens each frame) with varying shifted window sizes, and measure the time cost w/wo parallel generation. The effect of Swin attention equals to full attention when setting window size to 32. As shown in Figure 8, (1) Applying swin attention to auto-regressive generation accelerates inference. (2) Using parallel generation can further speed up inference without affecting generated videos, and achieves around 2× acceleration when window size ≤ 8.

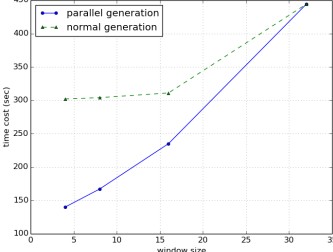

Figure 8: Acceleration results of parallel generation in autoregressive generation with Swin attention.

## B  ATTENTION ANALYSIS

To explore the attention mechanism of *dual-channel attention*, we visualize (1) the attention distribution in the temporal channel and (2) the mixture factor $\alpha$ controlling the ratio between the spatial and temporal channel in equation 1.

Figure 9 visualizes the distribution among frames and texts in sequential generation (Stage 1) with heat maps, where only 24 of 48 attention heads in 6 layers are shown for display purposes. The attention patterns can be broadly classified into the following categories:

- Most of the attention is on the text. E.g. the attention heads in violet.

- Most of the attention is on a certain frame. E.g. the attention heads in pink focus mainly on the previous frame; the attention heads in blue focus mainly on the first frame besides the text; the attention heads in yellow focus mostly on the frame itself.

- Attention is spread over several frames. E.g. the attention heads in green.

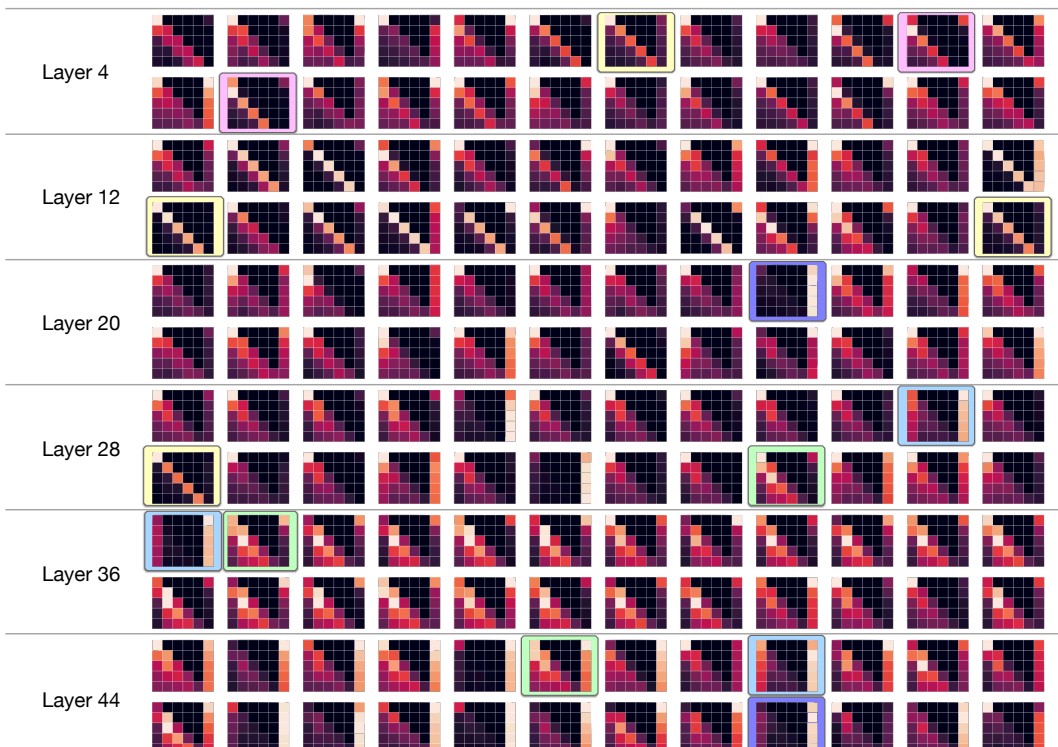

Figure 9: The attention distribution among frames and texts in sequential generation (Stage 1). Only 24 of 48 attention heads in 6 layers are selected for display purposes. Each attention head is visualized with a heat map of size 5×6, where lighter color represents a larger value. The 5×5 block on the left indicates the sum of attention scores (after softmax) between each pair of frames, and the rightmost column indicates the sum of the attention score of each frame to text. That is to say, the grid in row i column j ($j \leq 5$) represents $\sum_{x \in F_i, y \in F_j} \text{attn}_{x,y}$, and the grid in row i column 6 represents $\sum_{x \in F_i, y \in T} \text{attn}_{x,y}$, where $F_i$, $T$ denotes the set of tokens in the i-th frame and text respectively, and $\text{attn}_{x,y}$ denotes the attention score of token x to y.

Some attention heads exhibit a single pattern, while others may exhibit a mixture of them. Attention heads in the same layer tend to show similar patterns. In lower layers (e.g. layer 4, 12) the heads tend to allocate attention according to position, while in higher layers more attention is allocated to text (e.g. layer 44) or spread over multiple frames. One possible explanation is that there are more high-level features in higher layers such as video semantics, by which more frames and texts can interact with each other to make high-level feature analyses.

It is worth noting that many heads in the temporal channel do not allocate much attention to the frame itself, especially in higher layers, while attending to itself is important for inference. This shows that the CogVideo performs a certain degree of decoupling in the analysis of temporal and spatial features. While the spatial channel is in charge of feature analysis within the frame, the temporal channel can allocate more resources to explore relationships among different frames. We further illustrate this perspective with Figure 10, which shows that features calculated by CogView2 in the spatial channel are heavily relied on.

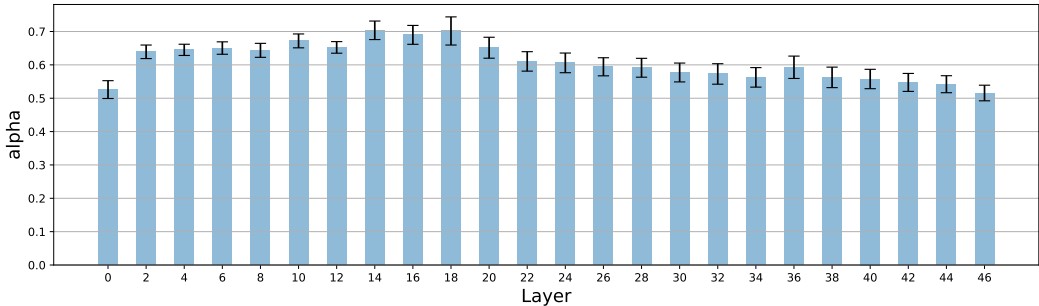

Figure 10: The mixture factor $\alpha$ controlling the ratio between the spatial and temporal channel in equation 1 in dual-channel attention. Only $\alpha$ in half of the layers are shown for display reasons. As $\alpha$ is a vector of dimension 3072, we show the mean and variance among all of its dimensions in this figure.

## C  DETAILS ABOUT HUMAN EVALUATION

In this section, we introduce more details about human evaluation for measuring generation quality. The conduction of our human evaluation generally follows previous works including Ramesh et al. (2021); Ding et al. (2021)

We randomly extract 30 classes from UCF101 for video generation, using corresponding video samples in the dataset as ground truth items in the evaluation. Based on captions of selected classes, we generate video samples from models including TGANv2, VideoGPT, and our model, CogVideo. To further illustrate the effectiveness of hierarchical multi-frame-rate generation, we also include a 1-stage version of the CogVideo model which only uses the Stage 1 model and extends samples by circularly reinserting the last frame into the model. For TGANv2, we use the official source code to train an unconditional generation model under the same setting as that in Saito et al. (2020). For VideoGPT, we use the official unconditional pretrained model to generate samples. To assign unconditionally generated samples into corresponding categories, we choose TSMLin et al. (2019) as the action recognition model for post-classification. We only keep the samples whose likelihood of a certain class is at least 80%. A randomly selected subset of samples is displayed in Figure 11.

For each sample of the video mentioned above, we ask evaluators to give scores between 1 and 5 ( 5 indicates the best while 1 indicates the worst) from three aspects including frame texture, motion realism, and semantic relevance. Then the evaluators are required to give a general score of quality for each sample between 1 and 10, where a higher score indicates better quality. After video samples from each caption are evaluated, the evaluators are asked to select the best one from them. We show snapshots of the evaluation website in Figure 12.

Throughout the process of human evaluation, we invited nearly 100 anonymous evaluators, while 90 of them completed the whole evaluation and were counted in the final results. None of the questions in the evaluation have any time limit. We offer each evaluator 75 RMB as a reward for the evaluation. Results of the human evaluation, including the average score and standard deviation for each group, have already been introduced in Figure 4 in the main body. As ground truth samples take an absolute predominance in the best selection question, we have removed the part of ground truth samples in the selection pie plot for clearer model comparison.

## D  GENERATED VIDEO SAMPLES

Thanks to the recursive interpolation model in Stage 2, CogVideo is able to generate relatively high-frame-rate videos, as shown in Figure 13. We provide further examples generated by CogVideo in Figure 14. *The generated videos in mp4 format can be found in supplementary material, with the filename "CogVideo_samples.mp4".* The length and the frame rate of provided videos are 4 seconds and 8 fps, respectively.

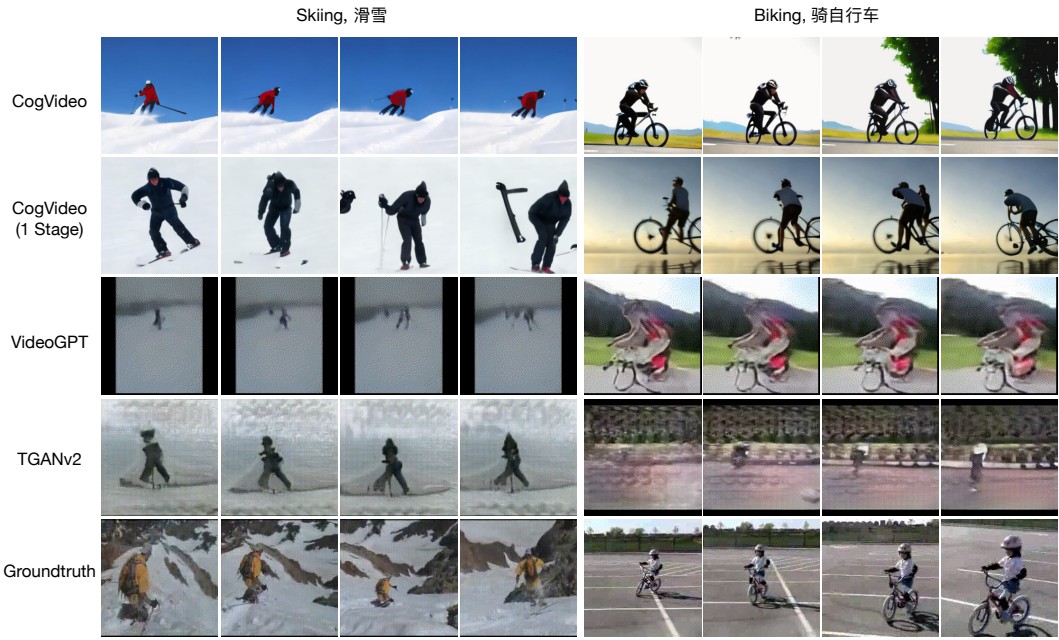

Figure 11: A subset of human evaluation samples. The captions are randomly selected from UCF-101. The original samples are clips of 16 frames, which are downsampled to 4 frames uniformly for display purposes.

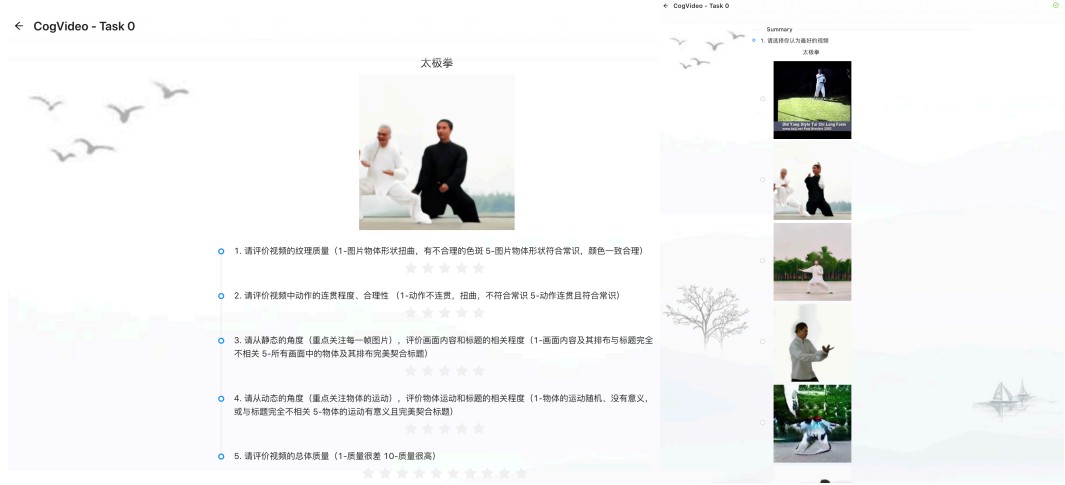

Figure 12: Snapshots of the evaluation website.

A man is running in the sea. 一个男人在海里跑步。

Figure 13: A 4-second video sample generated by CogVideo, which is firstly sequentially generated at 1 fps and then recursively interpolated for 3 iterations.

## E   SUPPLEMENTAL HUMAN EVALUATION AND QUALITATIVE COMPARISON

In order to demonstrate CogVideo's performance more thoroughly, we additionally conduct human evaluation to compare it with Video Diffusion Model (VDM)(Ho et al., 2022) and NUWA(Wu et al., 2021b). Considering both of them didn't release codes or checkpoints (which is the reason for not including them in our original human evaluation), we evaluate on the 28 text prompts shown on the webpage of VDM and 6 prompts on the webpage of NUWA. We invite 21 anonymous evaluators to rate on 3 aspects with score 1-5 (5 indicates the best): overall quality, frame texture and content, and motion realism. The results are shown in Figure 15, indicating CogVideo gets better scores on all three metrics.

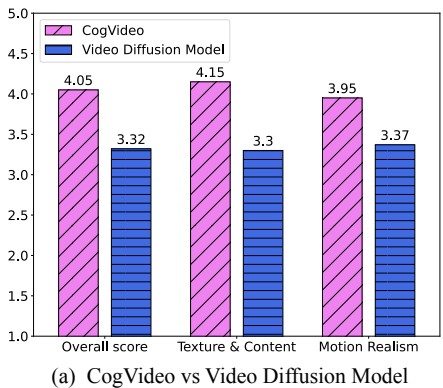
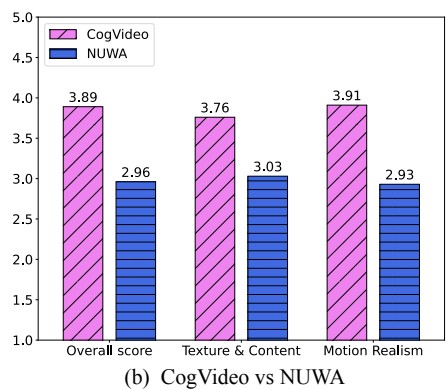

(a)  CogVideo vs Video Diffusion Model          (b)  CogVideo vs NUWA

Figure 15: Supplemental human evaluation. (left) CogVideo vs VDM; (right) CogVideo vs NUWA.

We further provide the qualitative comparison with NUWA(Figure 16) and Video Diffusion Model(Figure 17). From the samples we can see that, CogVideo can generate more realistic and detailed objects, e.g. "ducks in a pond"(VDM), "running on the sea"(NUWA), "a man is folding a piece of yellow paper"(NUWA). CogVideo can also generate videos with better motion realism and semantic alignment. For example, 1) in "pouring coffee into coffee cup"(VDM), the liquid level keeps raising in CogVideo's sample, while the liquid level sometimes drops in VDM's sample. 2) in "sunset at sea"(VDM), CogVideo shows the whole process of sunset.

## F    OUT OF DISTRIBUTION SAMPLES

While our training data only consists of real videos (rather than animations) and a small amount of abstract video (e.g. animation of abstract texture), CogVideo is capable of generating out-of-distribution (OOD) videos.

- Videos not existing in the real world, such as animals acting like human beings.
- Stylized videos such as watercolor painting and Chinese traditional drawing, while our dataset does not include any stylized videos.

Samples are shown in Figure 18. The out-of-distribution generation capability of CogVideo is two-folds:

- Frame-level: generate reasonable images not exisiting in the dataset. CogVideo losslessly inherits the OOD frame-level generation capability from CogView2 since it preserves all CogView2' parameters, showing the superiority of dual channel attention.
- Video-level: given OOD frames, CogVideo is able to generate reasonable actions. For example, when giving an image of a cat with red hat playing the guitar, CogVideo can transfer the human hands to cat's paws, and make it pluck the guitar strings.

We have to admit that, though, the success rate of extreme out-of-distribution generation is not very high, due to several reasons: 1) Generalization is too hard for extreme out-of-distribution cases, e.g. birds playing guitar. It's difficult to relate birds' wings to human arms. 2) The generalization capability is bounded by the image generation model, and sometimes even the first frame is in poor quality.

## G    LIMITATIONS

Although CogVideo demonstrates state-of-the-art performance in text-to-video generation, it still has certain limitations and sometimes produces failure cases. The major limitations are summarized below. We further attach each of them with possible solutions. It is worth noting that, because CogVideo is based on autoregressive stochastic sampling, these text prompts can mostly yield good cases with multiple times of sampling.

1. The quality heavily relies on the first frame, thus CogVideo inherits limitations from autoregressive text-to-image generative model, including:
   - A) Unreasonable shape, which may exist throughout the video.
   - B) Text-content mismatching, especially when the text is complex. E.g. missing component, component mismatch, incorrect attributes binding.
   - C) If there are flaws in the first frame, sometimes CogVideo is not robust enough to self-recover, or even amplifies the flaws.

   Increasing data and training for both the based image model (CogView2) and CogVideo may alleviate these problems. Leveraging pretrained text model can boost text understanding.
2. Slight temporal inconsistency (unnatural changes). Possible solutions include joint modeling multiple frames in super-resolution and increasing training time.
3. Slight blurry induced by VQ-VAE's lossy compression. Our $480 \times 480$ pixel results are decoded from $60 \times 60$ tokens with VQ-VAE, thus may contain slight blur. One possible solution is to further train a super-resolution or deblur model on the pixel level.

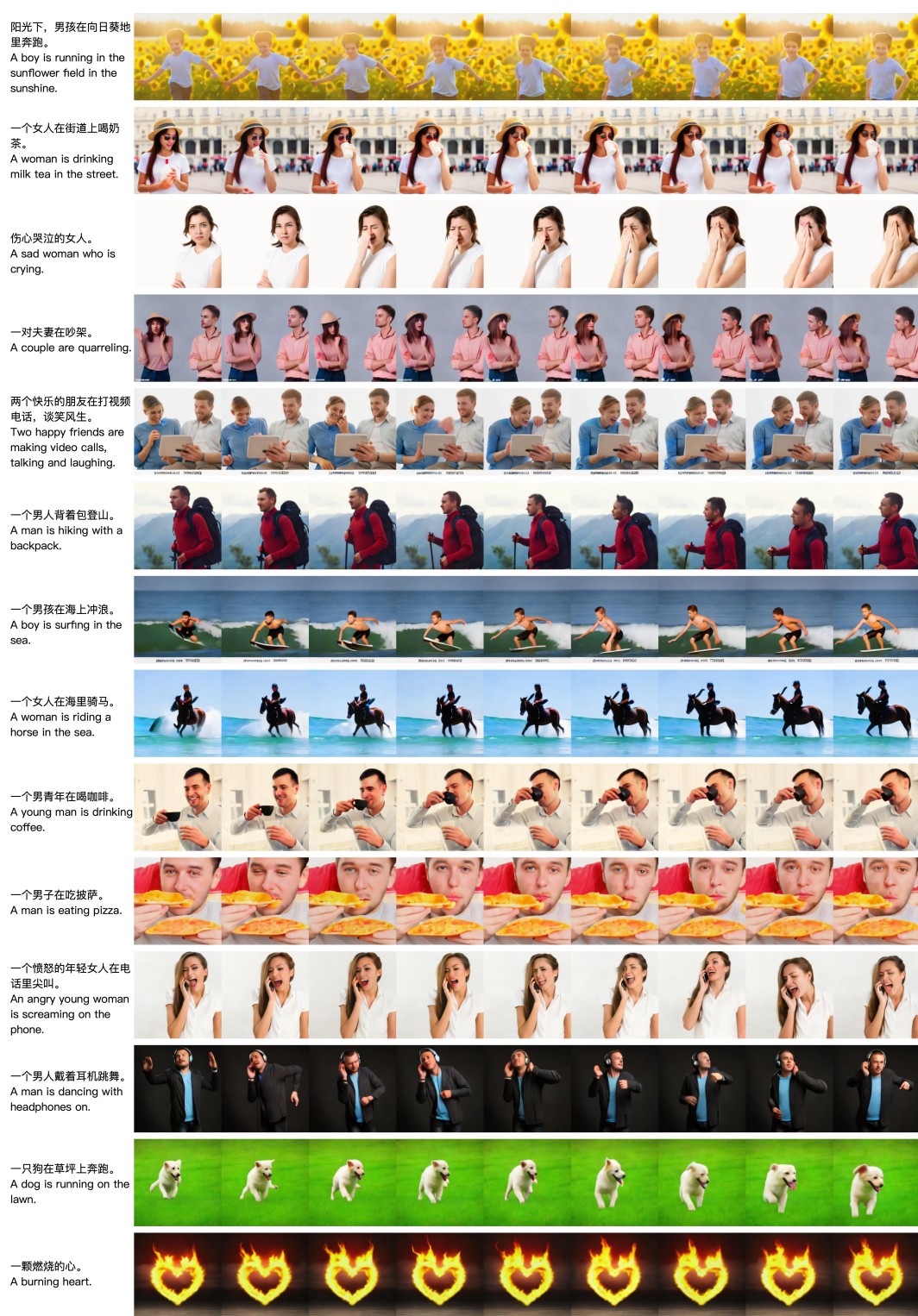

Figure 14: Further samples generated by CogVideo. The actual text inputs are in Chinese. Each sample is a 4-second clip of 32 frames, and here we sample 9 frames uniformly for display purposes.

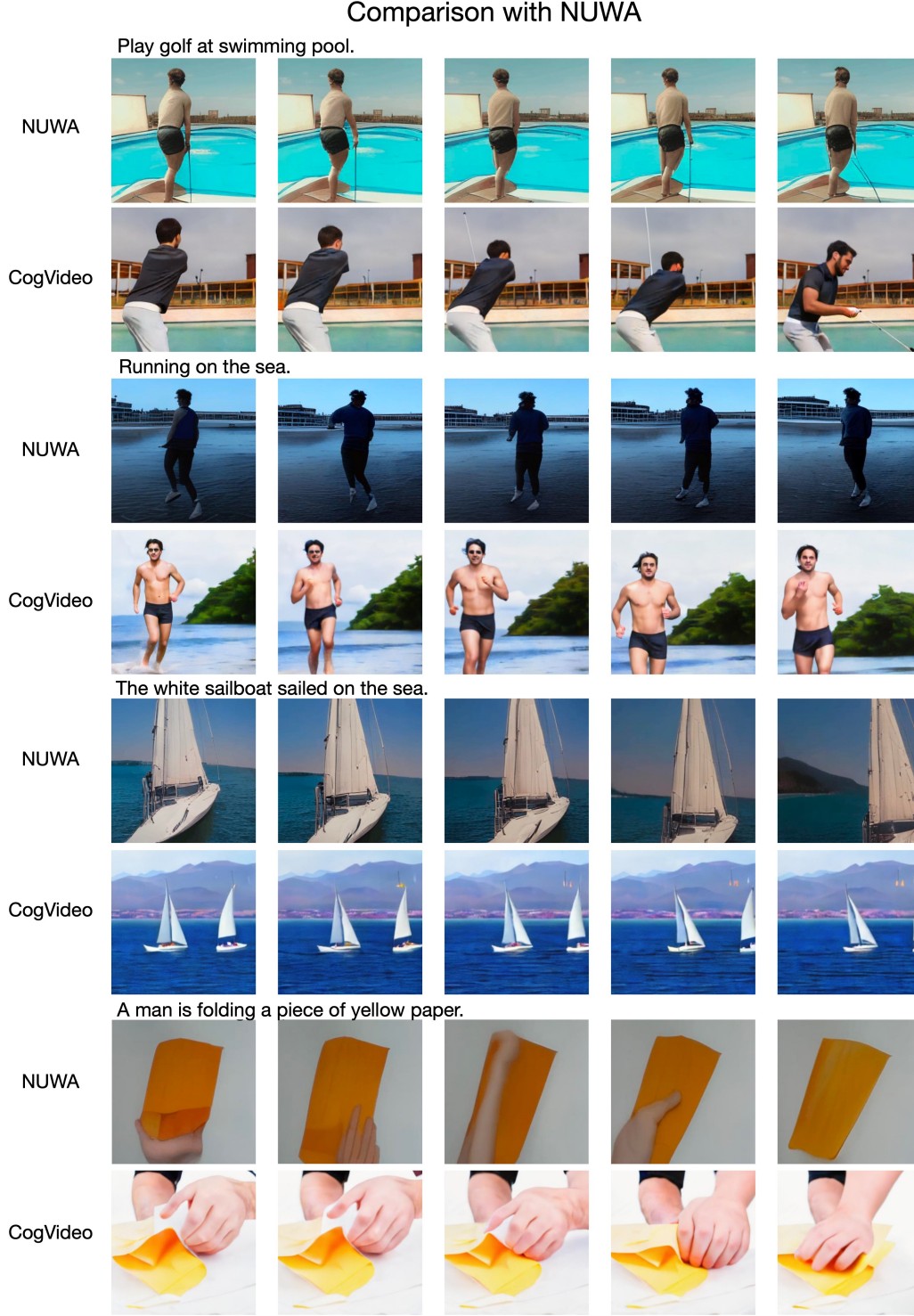

Figure 16: Qualitative comparison with NUWA(Wu et al., 2021b). Samples of NUWA are obtained from the paper's appendix. Samples of CogVideo generated by Stage 1 model, with frame rate of 1 fps.

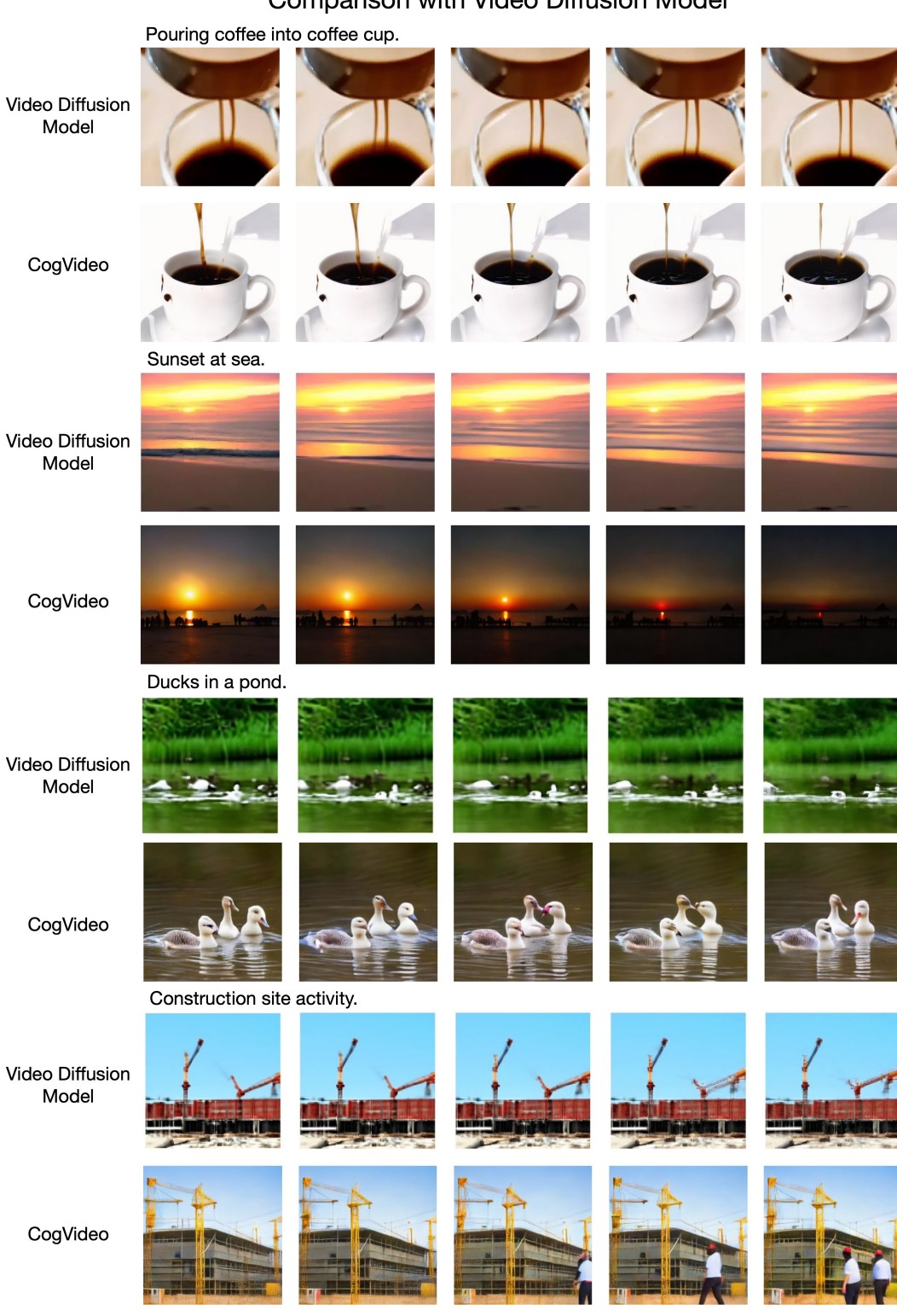

Figure 17: Qualitative comparison with Video Diffusion Model(Ho et al., 2022). Samples of Video Diffusion Model are obtained from their official website `https://video-diffusion.github.io`. Samples of CogVideo generated by Stage 1 model, with frame rate of 1 fps.

A cat in red jacket and blue jeans is dancing,

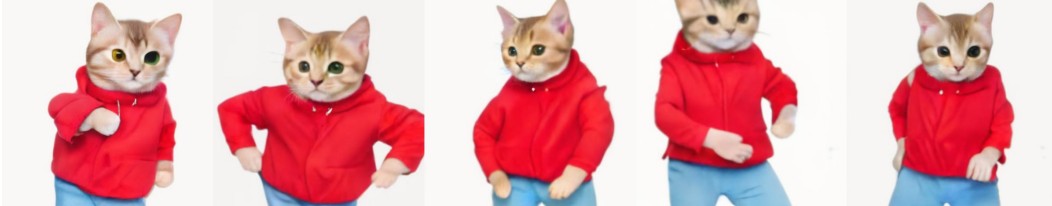

A cat in red jacket and blue sunglasses is drinking tea from a bowl.

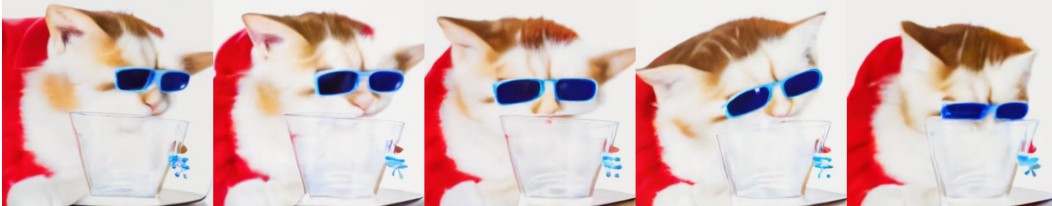

A cat in red hat is playing the guitar.

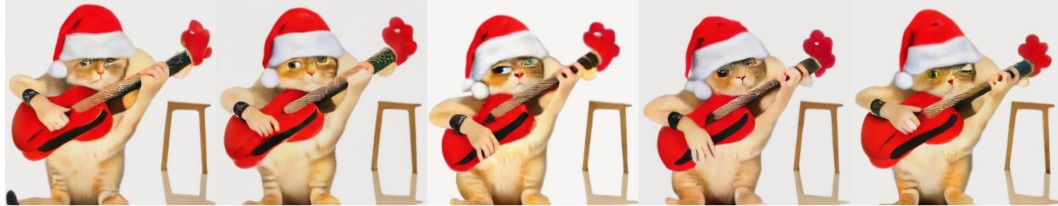

A cat is drawing on a drawing board.

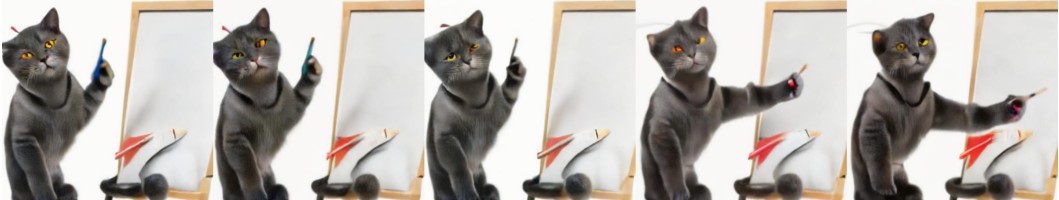

People walking in the rain with unbrellas, Chinese traditional drawing.

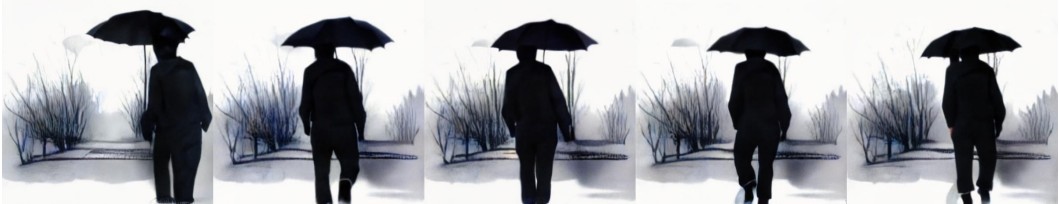

People walking in the rain with unbrellas, watercolor painting.

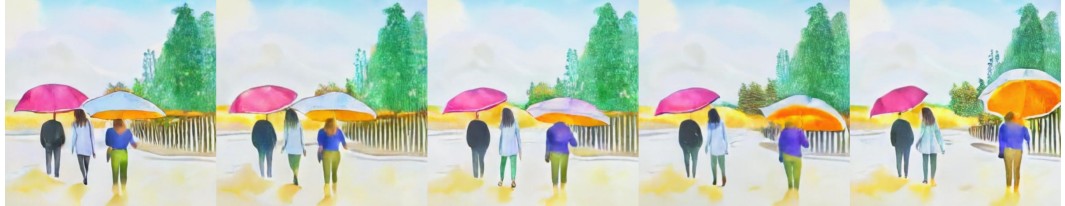

Figure 18: Out of distribution samples generated by CogVideo.

A. Shape (prompt: drawing on paper with a pen)

B. Text-content matching (prompt: a cat drinking a bottle of milk with a straw)

(prompt: a dog is typing on the keyboard)

C. Self-recovery (prompt: an oil painting of a couple in formal evening wear going home get caught in a heavy downpour with umbrellas)

Figure 19: Some failure cases generated by CogVideo.

