# OpenReview forum: "CogVideo: Large-scale Pretraining for Text-to-Video Generation via Transformers"
_ICLR.cc/2023/Conference — ICLR 2023 poster_

### Official Review · Reviewer_M4hz · 2022-10-23

**Confidence:** 4
**Correctness:** 3
**Technical Novelty And Significance:** 2
**Empirical Novelty And Significance:** 3
**Recommendation:** 6

**Clarity, Quality, Novelty And Reproducibility:**

The paper focuses on engineering improvements (which are valuable), hence the technical novelty is not signifiant. The paper is mostly well-written, with some arguments (e.g., motivation of multi-frame-rate training) not well-supported.

**Strength And Weaknesses:**

Strengths:
- Text-to-video generation is an emerging and exciting field. This paper is one of the first work to demonstrate the possibility of such a system.
- CogVideo achieves good performance on several benchmarks, outperforming previous methods by a large margin.
- The paper provides sufficient engineering details. The code and model is also open-sourced.

Weaknesses:

- Why must the videos be sampled into a fixed number of frames during training? This fixed-length restriction seems to be the fundamental motivation of the proposed multi-frame-rate training, but the reason of this restriction is not clear. An autoregressive transformer should be able to process variable-length input. I assume that the GPU memory constraint is the main reason? If so, there could exist better solutions as discussed below.

- The proposed method flattens a video as a sequence of spatial-temporal image patches. While this seems to be a straightforward extension from a text-to-image autoregressive model, it incurs a large amount of computation cost. One alternative solution is to encode each frame (instead of each patch) into a latent vector, and uses another model to generate the frame from the latent vector. Could the authors discuss on the pros and cons on these two design choices?

- Is there any positional encoding to indicate the spatial-temporal position of each patch?

- One potential concern for the current model architecture is that the video generation process may rely too much on the video inputs and less on the text, as text only occupies a small part of the input sequence. Have the authors considered using cross-attention on text instead?

- How important is text information for the second interpolation stage? Is it possible to train an interpolation model using only video data?

- The proposed multi-frame-rate training method does not fundamentally address the misalignment between text and video. The misalignment occurs due to the weakly-annotated dataset. CogVideo is still trained on web datasets. Have the authors performed any data cleaning to ensure better quality than Howto100M?


**Summary Of The Paper:**

This paper proposes an image-to-text generation model based on the autoregressive transformer architecture. The videos are flattened into a sequence of spatial-temporal patches, tokenized by a VQVAE, and concatenated with the text input. The authors propose several engineering improvements that lead to more effective training. The resulting CogVideo model outperforms previous methods on multiple benchmarks.

**Summary Of The Review:**

The biggest value of this paper is that it is one of the first text-to-video generation models with reasonable performance. The proposed engineering improvements could be useful. However, I have some concerns regarding the motivation of the proposed method and its advantage over alternative model designs. Overall, I'm slightly leaning towards acceptance, and may change my score depending on the authors' response.

---

> ### Author Response · Authors · 2022-11-18
> **Authors' response (2/2)**
>
> > **Weakness 5:** How important is text information for the second interpolation stage? Is it possible to train an interpolation model using only video data?
>
> For frame interpolation at a high frame rate, text information may not be a must because such information can be inferred from context frames. However, the Stage 2 of CogVideo requires interpolation at a low frame rate (e.g. from 1 fps to 2 fps), at which text may provide guidance.
>
> One example:  https://i.imgur.com/5ETWnkc.jpg . We provide ground-truth frames of "walking at the seaside" at the frame rate of 1 fps, and ask Stage 2 model to interpolate it to 2 fps with different text prompt ("walking at the seaside" or "running at the seaside").  The interpolated frames (marked with dotted line boxes) will be influenced by the text prompt.
>
> > **Weakness 6:** The proposed multi-frame-rate training method does not fundamentally address the misalignment between text and video. The misalignment occurs due to the weakly-annotated dataset. CogVideo is still trained on web datasets. Have the authors performed any data cleaning to ensure better quality than Howto100M?
>
> TLDR: Regarding the quality and format of current text-video datasets such as WebVid-10M, annotation quality is no longer significant bottleneck. Therefore, multi-frame-rate training can greatly address the misalignment problem and boost semantic matching.
>
> Recently, video datasets with relatively high-quality annotation can be found on the Internet, e.g. WebVid-10M, whose annotation quality is much higher than Howto100M (samples can be found at https://m-bain.github.io/webvid-dataset/). Our dataset is of similar format and quality to WebVid-10M, but with a slightly smaller scale (5.4M vs 10M).  **Hence, annotation quality is no longer significant bottleneck for text-video alignment.** However, one problem is that each video is usually paired with only 1 caption describing the whole video in these datasets, despite their varying lengths (ranging from 1 to 60 sec in our dataset), thus prevailing fixed-frame-rate sampling will destroy text-video alignment. **This problem is particularly prominent in pretraining**, because the demands large-scale datasets are usually widely collected from the Internet and their video lengths vary a lot. **Therefore, we propose multi-frame-rate training.** **This technique can be extended to various generative models/pipelines.** For example, Make-A-Video, the concurrent video generation model based on diffusion model, applies this technic in their pretraining.

---

> ### Author Response · Authors · 2022-11-18
> **Authors' response (1/2)**
>
> Thanks for your careful and valuable review! Guided by your insightful questions, we re-clarify our motivation and provide deeper insight into the advantages of our technical designs (with some supplemented experiments). We will address your concerns point by point as follows:
>
> > **Weakness 1:** Why must the videos be sampled into a fixed number of frames during training?
>
> Thanks for the insightful question! The reasons are twofold, considering the characteristics of both pretraining and video generation. To the best of our knowledge, nearly all previous video generation works adopt this strategy, such as VideoGPT, Video Diffusion Model, etc.
>
> 1) Computational efficiency, which is critical for pretraining. Since we use mini-batch gradient descent, each sample should have the same length within a batch. If we fix the frame rate and unfix the number of frames, videos of shorter duration should be padded to align with the longest one in the batch, resulting in huge time and energy waste. That's why pretrain models such as GPT3, PaLM cut articles to sequences of the same lengths.
> 2) Difficult to balance the frame rate and maximum duration. Due to GPU memory constraints, one sample can only accommodate a small number of frames (refer to *the answer to weakness 2*), but the video duration varies a lot. On the one hand, if we want to contain videos of long duration, an extremely low frame rate should be used (some short videos may even be sampled into 1~2 frames). On the other hand, if a high frame rate is used, long videos have to be clipped, resulting in text-video misalignment.
>
> > **Weakness 2:** . One alternative solution is to encode each frame (instead of each patch) into a latent vector, and uses another model to generate the frame from the latent vector. Could the authors discuss on the pros and cons on these two design choices?
>
> **Although in video understanding tasks we can represent each frame with a single vector, in generative tasks, nearly all the current work encode frames at patch-level, because encoding with a single vector will cause intolerable loss of details in frames.**
>
> According to our understanding, the essence of the alternative solution is to trade off image quality against memory consumption (number of latent vectors). While frame quality is critical to a video's visual effect, a single latent vector is impractical to encode sufficient details of a frame. Here we show the result of a supplementary experiment: we trained VQ-VAE of different compression rates (compress 256\*256 pixel image into 64\*64 or 32\*32 or 16*16 vectors). The reconstruction results are shown in https://i.imgur.com/aGaDHW7.jpg . Even 16\*16-vector compression shows strong blur and artifacts. To the best of our knowledge, nearly all SOTA transformer-based video/image generation models are based on patch-level encoding, e.g. NUWA, VideoGPT, TATS. **Therefore, patch-level encoding seems to be a necessary implementation, rather than being constrained by text-to-image models.**
>
> > **Weakness 3:** Is there any positional encoding to indicate the spatial-temporal position of each patch?
>
> Yes. We adopt the standard transformer implementation. (Learnable) position embeddings are added to each hidden vector before the first layer.
>
> > **Weakness 4:** One potential concern for the current model architecture is that the video generation process may rely too much on the video inputs and less on the text, as text only occupies a small part of the input sequence. Have the authors considered using cross-attention on text instead?
>
> Similar concerns have arisen before in text-to-image generation, where text tokens are far less than image tokens. Based on the results of text-to-image/video models e.g. DALL-E, NUWA, models can successfully learn to capture the relationship between text and image when sharing the same attention for text and image tokens. Therefore, CogVideo borrowed the design of those models for simplicity. Our generative results also suggest that this design successfully captures text-image relationships. In Appendix B we further investigate the attention mechanism, and find that CogVideo automatically learns multiple types of attention heads, and some of them allocate most attention to text (Figure 9).

---

> > ### Comment · Reviewer_M4hz · 2022-11-20
> > **Image-level vs. patch-level encoding.**
> >
> > Thanks for the response! For weakness 2, I am not fully convinced that "patch-level encoding seems to be a necessary implementation", as stable-diffusion and LDM uses a VAE for image compression and demonstrate good results.

---

> > > ### Author Response · Authors · 2022-11-20
> > > **Response about "Image-level" and "patch-level" encoding**
> > >
> > > Thank you very much for your quick response. However, stable (latent) diffusion [1] also uses "patch-level" encoding in our opinion. To make sure that we understand your question correctly, please allow us to first synchronize some background definitions with you:
> > >
> > > **Background (Encoding methods)**
> > >
> > > We can roughly divide the methods into three categories, pixel-level, patch-level, and image-level, based on the modeling space.
> > >
> > > 1. **pixel-level**, including ImageGPT [2], Imagen, MAE (understanding) and many GANs. To directly generate pixels, we need to work on a very large resolution, which is unaffordable for Transformers, even though Unet is possible. VQ-VAE [3] solved this problem via **patch-level** encoding.
> > > 2. **patch-level**, including DALL-E, CogView, latent diffusion [1], etc. These methods are based on KL-reg VAE or VQ-VAE [3]. They compress an image of $H \times W \times 3$ into $(H / f) \times (W/f) \times d$ by training an auto-encoder, which $f$ is the patch size (compression rate) and $d$ is the dimension of the latent space.
> > >    The $f$ cannot be very large, as we mentioned in the response. If $f>8$, there will be obvious artifacts when decoding.
> > >    In stable diffusion, $f=8, d=4$ means that they compress the $512\times 512\times 3$ into $64\times 64\times 4$. In CogVideo first stage, we use $f=8, d=\text{discrete}$ to compress $160\times 160 \times 3$ into $20\times 20 \times 1$. The latent diffusion paper [1] (quick link https://arxiv.org/pdf/2112.10752.pdf , Page 21, Table 8) also showed that the discrete encoding **performs similar** to the continuous encoding used in stable diffusion.
> > > 3. **image-level**, as you mentioned, compresses an image to a *single latent vector*. This is equivalent to a very large $f$ in the patch-level encoding.
> > >
> > > **Response to your question**
> > >
> > > Your question is "One alternative solution is to encode each frame (instead of each patch) into a latent vector, and uses another model to generate the frame from the latent vector. Could the authors discuss on the pros and cons on these two design choices?"
> > >
> > > In our understanding, if the "another model" refers to a decoder in VQVAE (image-level above), just as stable diffusion does but with a $f=160$, this hardly works because the heavy artifacts of reconstruction for a large $f$.
> > >
> > > If the "another model" is another **generative** model, this is indeed possible. The most close work might be DALLE-2 [4], which compress the image into the single CLIP image embedding vector and train another diffusion model to generate the image conditioning on this embedding. However for this method, the "another model" consumes more resources both in training and inference. This is a hierarchical solution, and our generation and super-resolution (text => 160 => 480) stage also shares the same spirit.
> > >
> > > If there are any misunderstanding, or you want more detailed discussion on this point, please tell us.
> > >
> > > [1] Rombach, Robin, et al. "High-resolution image synthesis with latent diffusion models." *Proceedings of the IEEE/CVF Conference on Computer Vision and Pattern Recognition*. 2022.
> > >
> > > [2] Chen, Mark, et al. "Generative pretraining from pixels." *International conference on machine learning*. PMLR, 2020.
> > >
> > > [3] Van Den Oord, Aaron, and Oriol Vinyals. "Neural discrete representation learning." *Advances in neural information processing systems* 30 (2017).
> > >
> > > [4] Ramesh, Aditya, et al. "Hierarchical text-conditional image generation with clip latents." *arXiv preprint arXiv:2204.06125* (2022).

---

> > > > ### Comment · Reviewer_M4hz · 2022-11-21
> > > > **Thanks for the clarification**
> > > >
> > > > Thanks for the response. This point is clear to me now.

---

### Official Review · Reviewer_5tA9 · 2022-10-25

**Confidence:** 3
**Correctness:** 3
**Technical Novelty And Significance:** 2
**Empirical Novelty And Significance:** 2
**Recommendation:** 3

**Clarity, Quality, Novelty And Reproducibility:**

Clarity: Good.

Quality: Neutral. The performance of the model is not so good.

Novelty: Limited. All of the proposed modules are existing techniques and are not very interesting.

Repoducibility: Neutral. There is open-source code released. But the data is still private.

**Strength And Weaknesses:**

Strengths:
1) The paper presents a large-scale video pertaining network.
2) The paper is well-written and easy to follow.

Weaknesses:
1) The results are not of high quality. From the anonymous web demo, the generated videos are blurry and not temporally consistent.
2) The novelty is limited. The whole architecture is just a combination of two transformers, one for sequential generation and the other for recursive interpolation. Both of them are existing techniques. They are directly adapted from VQVAE-based image generation methods.
3) The authors employ a temporal channel attention block to handle the temporal coherence. However, there is no ablation study on this. The authors only did ablation studies on fully fine-tuning them, partially fine-tuning them, or training from scratch. One possible ablation study is removing this module.
4) The comparisons on the UCF-101 and Kinetics-600 datasets are not fairly as the dataset used in CogVideo is significantly larger than others.
5) The improvements are mainly contributed to the 5.4 million text-video pairs. However, the authors seem not to have a plan to release the dataset. I understand the main focus of this paper is to provide a large-scale pertaining video generation network. However, the performance of this model is far from existing text-to-image models. Therefore, with this model but without data, other researchers still cannot follow this direction.
6) From Fig. 5, it seems that the results of (a) and (b) are both reasonable.
7) There is no qualitative comparisons with existing methods in the main paper.

**Summary Of The Paper:**

The paper presents a large-scale video pretraining network, i.e., CogVideo, for text-to-video generation. The results are plausible. The technical contributions of the paper are 1) Multi-frame-rate training, and 2) dual-channel attention.

**Summary Of The Review:**

The performance of the model is not so impressive. The proposed methods are not so interesting.

---

> ### Author Response · Authors · 2022-11-18
> **Authors' response (3/3)**
>
> > **Weakness 3:** The authors employ a temporal channel attention block to handle the temporal coherence. However, there is no ablation study on this. The authors only did ablation studies on fully fine-tuning them, partially fine-tuning them, or training from scratch. One possible ablation study is removing this module.
>
> "Removing this module"  equals to "naïve full parameter finetune (initialize with CogView2 and apply full parameter finetuning)" in our article. Sorry for the insufficient explanation and we've revised it in the pdf revision.
>
> > **Weakness 4:** The comparisons on the UCF-101 and Kinetics-600 datasets are not fairly as the dataset used in CogVideo is significantly larger than others.
>
> CogVideo is based on pretraining and thus inevitably uses a large dataset. Therefore, we follow previous pretraining works by finetuning and evaluating them on downstream tasks. This is a common practice in pretraining, e.g. BERT is pretrained on 3,300M words and evaluated on much smaller datasets.
>
> > **Weakness 5:** The improvements are mainly contributed to the 5.4 million text-video pairs. However, the authors seem not to have a plan to release the dataset. Therefore, with this model but without data, other researchers still cannot follow this direction.
>
> A good alternative solution is to use WebVid-10M for reproducing. We are sorry that due to copyright issues we can't release the dataset. However, the content and quality of our dataset are very similar to WebVid-10M (https://m-bain.github.io/webvid-dataset/), a recently released general domain dataset with 10M text-video pairs, thus one can refer to WebVid-10M to reproduce CogVideo (may yield better performance because WebVid-10M is bigger). WebVid-10M was not released at the time we started this project, therefore we used personally collected datasets.
>
> > **Weakness 6:** From Fig. 5, it seems that the results of (a) and (b) are both reasonable
>
> As you have suggested, both (a) and (b) are much more reasonable than (c), verifying the effectiveness of inheriting CogView. But (b) has relatively poorer quality than (a) at a close look. For example, in the case of "sled dogs running in the snow",  in (b) frame 3-5, the appearance of sled dogs becomes severely twisted; in the case of "a lion is dancing at the seaside", in (b) frame 2-5, the legs are in strange shapes and the number of legs keeps changing.
>
> > **Weakness 7:** There is no qualitative comparisons with existing methods in the main paper.
>
> Due to the space limitation of the main text, we showcased qualitative comparisons with VideoGPT and TGANv2 in Appendix (Fig. 11). Please check at our general response for further qualitative comparison with NUWA and Video Diffusion Model.

---

> ### Author Response · Authors · 2022-11-18
> **Authors' response (2/3)**
>
> > **Weakness 2:** The novelty is limited. The whole architecture is just a combination of two transformers, one for sequential generation and the other for recursive interpolation. Both of them are existing techniques. They are directly adapted from VQVAE-based image generation methods.
>
> This might be a misunderstanding. The techniques behind the proposed CogVideo are not that straightforward, and some of them are especially helpful for text-video pretraining. We have carefully tested each of them and found, without each of them, the video generation performance will be significantly reduced. To summarize, we will stress our four main contributions and why they are important.
>
> - **Contribution 1**: Using *Dual-channel Attention*, CogVideo elegantly and effectively finetunes text-to-video generative model from a pretrained text-to-image transformer.
>
>   Pretraining a large text-to-video model in the general domain suffers from a severe lack of data. For image generation, there are datasets with billions of image-text pairs (e.g. LAION-5B) while the text-video datasets are substantially smaller (e.g. WebVid ~10M videos), which is not enough given the higher complexity of open domain videos.
>
>   We propose *dual-channel attention* as a universal approach to efficiently inherit knowledge from pretrained image models without catastrophic forgetting, which not only **alleviates data scarcity** but also **greatly reduces computational resources (50% less optimized parameters, and reutilizing spent compute)**. According to the ablation study, it shows **improvement both quantitatively (FVD by 40%) and qualitatively**. It can also be applied in other tasks and modalities, e.g. finetuning an image generative model to a super-resolution model, etc.
>
> - **Contribution 2:** *Multi-frame-rate training* to better align text-clip pairs, which significantly improves the generation accuracy, in particular for movements of complex semantics.
>
>   The semantic alignment between text and video is the key to text-to-video generation. However, pre-training poses a new challenge for this: Since datasets for pretraining are usually widely collected from the real world, the videos' lengths vary a lot. Previous fixed-frame-rate strategy requires cutting videos into sub-clips of the same length and forcing the caption to pair with these sub-clips, resulting in misalignment.
>
>   By contrast, *multi-frame-rate training* models the video together with the frame rate, which simply yet effectively **solves the misalignment problem**. **We demonstrate its effectiveness in the ablation study (sec 5.3.2**). In addition, it offers CogVideo **the capacity of controlling the intensity of changes** during generation.
>
> - **Contribution 3**: Adopting Shifted-window attention to the auto-regressive scenario, CogVideo accelerates both training and inference.
>
>   A prominent problem for text-to-video pretraining is the huge time and memory overhead during both training and inference, attributed to the large model and long sequence. By auto-regressive swin attention, CogVideo is able to conduct parallel generation in faraway regions of different frames, and **achieves around 2× acceleration** when window size ≤ 8 (See Appendix A).
>
> - **Contribution 4**: CogVideo performs text-to-video generation in the general domain.
>
>   As the **largest** pretrained transformer for text-to-video generation, CogVideo is able to **work in different domains** including but not limited to people, animals, objects, landscapes, etc. In comparison, most existing SOTA models such as DIGAN, VideoGPT, and TATS can only work for a specific domain. We have open-sourced the code and pretrained model to benefit the development of this topic.

---

> ### Author Response · Authors · 2022-11-18
> **Authors' response (1/3)**
>
> Thanks for your careful review, and we will address your concerns point by point and clarify a few potential misunderstandings as follows:
>
> > **Weakness 1:** The results are not of high quality. From the anonymous web demo, the generated videos are blurry and not temporally consistent.
>
> By experiment, we show that CogVideo **outperforms most publicly available models by a large margin in human evaluation** and **demonstrates state-of-the-art performance in machine evaluation**. Please check at our general response for a newly added **human evaluation with close-sourced SOTA, Video Diffusion Model (2022) and NUWA(2021)**, where CogVideo yield the **best overall scores**.  As **one of the first pretrained text-to-video generation models with reasonable performance** (*reviewer M4hz*), CogVideo inevitably has some deficiencies and we attach a chapter describing major limitations in Appendix.
>
> A general introduction to text-to-video generation: As mentioned in sec 2, most earlier works (e.g. VideoGPT(2021), DIGAN(2021), TATS(2022)) can only generate videos in a single domain, such as the changes of sky, TaiChi. And their performances are far from satisfactory (see UCF-101 samples of VideoGPT at https://wilson1yan.github.io/videogpt/index.html), with objects twisted or even hard to recognize. Furthermore, most of them have resolutions no higher than 256\*256 (while CogVideo is 480\*480). Recently, a few pretrained text-to-video models emerges and achieved certain improvements, e.g. NUWA(2021) (resolution 256*256), Video Diffusion Model (2022) (resolution 128\*128). From the evaluation results and case study in our general response, we show that CogVideo has advantages in overall quality, shape and semantic relevance.
>
> In conclusion, we hope that CogVideo is a meaningful initial step towards better video generation models. In the meantime, We are trying to improve the performance (e.g. blurry, temporal consistency as you have mentioned) in our next version, and have open-sourced our codes and checkpoints to boost the community.

---

### Official Review · Reviewer_kSBX · 2022-10-25

**Confidence:** 3
**Correctness:** 4
**Technical Novelty And Significance:** 3
**Empirical Novelty And Significance:** 2
**Recommendation:** 8

**Clarity, Quality, Novelty And Reproducibility:**

The paper is very well written and the quality and novelty of the work is very high. Regarding reproducibility, it seems that the authors have collected their own dataset, so unless they release it, it's going to be hard to reproduce the results from the paper. The proposed methods however, should be relatively easy to implement and tested in other datasets, given the details provided by the authors.

**Strength And Weaknesses:**

**Strengths**
- The paper presents a thorough evaluation of their model, based on both automatic scoring (using FVD and Inception scores), as well as a human evaluation from 90 anonymous evaluators (that were economically. compensated for their work).
- Some of the key (according to the authors) components were carefully ablated, i.e. the effectiveness of dual-channel attention.
- The appendinx contains very useful details about the implementatin and the evaluation of the model. I was able to find the answer to many questions that I had in these, especially regarding human evaluation. Kudos to the authors.

**Weaknesses**
- Several ways of "factorizing" the Temporal and Spatial attention have been presented in the past, but the paper does not compare against these approaches. For example, see the factorized attention in the "ViViT: A Video Vision Transformer" paper or in "VidTr: Video Transformer Without Convolutions".
- The effectiveness of the multi-frame-rate training is not ablated in the paper, which is a central contribution of the paper, according to the authors, there's merely a visual inspection of the type of videos generated conditioned on different frame rates (see figure 6).


**Summary Of The Paper:**

The paper presents an autoregressive model for text-to-video generation. The model is based on a Transformer backbone, pre-trained on a text-to-image task, and has three key components: 1) multi-frame-rate training (i.e. during training, the model observes videos sampled at different frame rates so that it can capture both short and long-term context); 2) a frame interpolation model to insert transition frames; and 3) the use of dual-channel attention (i.e. spatial and temporal attention is done in two separate but parallel layers, and the outputs are interpolated with a learnable parameter). Both the machine evaluation (using the popular FVD and Inception scores) and human evaluation show that the proposed approach produces better quality videos than the considered baselines (7 baselines, several of which published in the current year, 2022, were considered).

**Summary Of The Review:**

I believe that the paper could have good chances to influence the community, except for the fact that several industrial labs have recently published text-to-video image generation models that seem to generate higher quality videos (although this is of course debatable, since there isn't an apples-to-apples comparison available). Regardless of this, I believe that the work is of high quality, the experimentation and evaluation is sound, and it should be accepted to the conference.

---

> ### Author Response · Authors · 2022-11-18
> **Authors' response**
>
> Thanks for your careful and valuable review, and we will address your concerns point by point as follows:
>
> > **Weakness1:** Several ways of "factorizing" the Temporal and Spatial attention have been presented in the past, but the paper does not compare against these approaches. For example, see the factorized attention in the "ViViT: A Video Vision Transformer" paper or in "VidTr: Video Transformer Without Convolutions".
>
> Thanks for your insightful questions! Factorizing temporal and spatial attention has been applied in some Transformer-based video models for computational efficiency. However, our dual-channel attention differs from those works in the following aspects:
>
> 1. Adjusted for video generation, rather than video classification;
> 2. Efficiently and effectively inherit CogView2, without tuning FFN and spatial attention channel.
>
> To put it in detail: *ViViT* and *VidTr* discussed multiple types of spatial-temporal attention factorization, which cover the general structure of most existing factorization and are very enlightening. Here we discuss them one by one, and explain the reason or advantage of using dual-channel attention.
>
> - *Factorised Encoder* divides the model into two separate transformer encoders. Though being simple, it lacks the modeling of complex temporal-spatial relationship at multiple levels, which may work for video understanding but fail on more complex tasks such as video generation.
> - *Factorised self-attention* computes self-attention spatially and temporally in turn, which is very similar to *VidTr* (except that VidTr additionally uses a temporal down-sampling method to remove temporal redundancy, which is suitable for understanding tasks). However, since we want to freeze FFN (containing 50% parameters of CogView2) during training, the output feature space of factorized attention should be the same as CogView2's attention block. Intuitively, serially applying two attention blocks would easily change the feature space, making learning harder.
> - *Factorised dot-product attention* is similar to our dual-channel attention. It works in bi-directional attention settings. However, if applied in uni-directional attention scenario such as auto-regressive generation, the tokens in later frames can not see the tokens spatially after them in former frames. Therefore, we apply 3D swin attention in temporal channel. It can also be seen as a generalization of temporal-only dimension in ViViT (with window size 1\*1\*T), thus flexibly adjusting model capability.  In addition, we weighted sum temporal and spatial attention rather than concatenating and applying linear transformation, which is more efficient while maintaining the same model capability.
>
> > **Weakness 2:** The effectiveness of the multi-frame-rate training is not ablated in the paper, which is a central contribution of the paper, according to the authors, there's merely a visual inspection of the type of videos generated conditioned on different frame rates (see figure 6).
>
> Thanks for your suggestion! Due to the lack of quantitative indicators and the high training cost of the model, we are not able to do a thorough ablation on multi-frame-rate training. However, we strive to evaluate multi-frame-rate on two aspects:
>
> 1. Qualitative results in sec 5.3.2, which demonstrate that the multi-frame-rate training strategy not only enables CogVideo to generate videos of multiple frame rates, but also helps better align texts and videos of varying lengths.
> 2. (Additional) Quantitative evaluation.  we additionally evaluate the perplexities of ground-truth videos of various frame rates when conditioning on another frame rate (averaged on 1024 samples). The results suggest that ground-truth samples get the lowest perplexity when conditioned on matched frame rate, and the perplexities become slightly higher when the difference between the conditioning frame rate and ground-truth frame rate becomes larger.
>
> |                            | Ground-Truth: 1 fps | Ground-Truth: 2 fps | Ground-Truth: 4 fps |
> | :------------------------: | :-----------------: | :-----------------: | :-----------------: |
> | **Model Condition: 1 fps** |       45.878        |       30.008        |       21.050        |
> | **Model Condition: 2 fps** |       46.201        |       29.992        |       20.822        |
> | **Model Condition: 4 fps** |       46.340        |       30.123        |       20.780        |
>
>
> > **Reproducibility:** Please refer to our general response and pdf revision for a supplemented reproducibility statement.

---

> > ### Comment · Reviewer_kSBX · 2022-11-20
> > **Thanks for your response**
> >
> > Thank you for your detailed comparison between the different factorization strategies. This should be summarized (I believe), in the main text, perhaps in the "related works" section, though.
> >
> > I still believe that the contribution of multi-frame-rate should be properly ablated, and that the reported qualitative (in the paper) and quantitative (in the authors' response) is somehow insufficient.
> >
> > I will keep my current score (8). From my point of view, this is a good paper. Congratulations to the authors.

---

### Official Review · Reviewer_9Xh6 · 2022-10-29

**Confidence:** 5
**Correctness:** 3
**Technical Novelty And Significance:** 3
**Empirical Novelty And Significance:** 2
**Recommendation:** 6

**Clarity, Quality, Novelty And Reproducibility:**

- Clarity: The paper is mostly clear however some sections can be improved such as 3.2.
- Quality: The quality of the provided results are great however it's not clear how the model will perform on new combinations.
- Novelty: the provided method, particularly fps conditioning is new to the best of my knowledge.
- Reproducibility: While the authors provided the source code, the paper lacks details of the model and the used data is private which severely affects the reproducibility.

**Strength And Weaknesses:**

===== Strengths
+ The paper is well motivated. The authors enumerate the main issues that make text to video a hard problem and try to address them one by one.
+ The evaluations are done using both human metrics and machine metrics (FVD).
+ The authors open sources the code behind the paper which is always a plus.
+ The authors provided a large number of videos in their demo website which is also appreciated.

===== Weaknesses
- Weak baselines: The main baselines for the paper (in human evaluation) is TGAN (published in 2017) and VideoGPT (2021). The first one is out-dated at this point and the latter generates videos at a low resolution (64x64 and 128x128) which can affect the human judgement. It's not clear if the authors reduced the resolution of their model for a more fair comparison. I also encourage the authors to provide a similar comparison with newer and higher resolution models such as  NUWA and Video Diffusion.
- Other problems: following the previous weakness and given the fact that there are not that many text conditional video generation models out there, I suggest authors to compare their model with in other video problems (such as frame conditional video prediction) where there are more baselines to compare to.
- Out of distribution results: While the videos generated by the model look impressive, it is not clear how the model performs for out of distribution queries. The paper only contains two "new compositions" that probably don't exist in the training set: "A lion man is drinking water" and "A tiger is playing football" while the rest of the examples probably exist in the training set. The demo website contains more such new compositions however the generated videos are not generate suggesting that the model is overfitted to the training data and is just playing videos from memory! Unfortunately, the training data is also private which makes such analysis even harder but I suggest authors to perform an analysis on this matter by generating examples which are more detailed and most likely non-existent in the training set (e.g. "a cat in red jacket and blue sunglasses is drinking tea from a bowl").
- Reproducibility: While the authors provided the source code, the paper lacks details of the model and the used data is private which severely affects the reproducibility. The authors also did not provide any numbers that indicate the required computation to train a 9B model. This is interesting given the fact that the paper mentions computational cost as one of the main issues in text to video.
- Writing. The quality of the writing can be improved. Particularly, I found the Section 3.2. to be poorly written, assuming that the reader is familiar with details of the previous work CogView2 and CogLM.



**Summary Of The Paper:**

CogVideo proposes a combination of models that can generate videos from a given description. The system uses two model that generate frames in a hierarchical way: the first model generates the "key-frames" while the other model interpolates between the generated key-frames. The authors also proposed "a dual-attention mechanism" which allows for mix video-image training. The model is trained on a non-public dataset of ~5-4M text-video pairs. The authors tested the quality of generated videos on Kinetics-600 and UCF-101 and also provided a human study on videos generated on random classes of UCF.

**Summary Of The Review:**

Overall, while the provided results look good, more analysis and comparison is required to show case the strengths and limitations of the model. Given more analysis the paper should have more impact on the field.

---

> ### Author Response · Authors · 2022-11-18
> **Authors' response**
>
> Thanks for your careful and valuable review! We will address your concerns point by point as follows:
>
> > **Weakness 1:**
>
> **Part1:** Weak baseline in human evaluation
>
> **Please check at our general response for a newly added human evaluation with Video Diffusion Model (VDM)  (all 28 text prompts)  and NUWA  (6 text prompts) , where CogVideo get better scores on all 3 aspects.**
>
> Unfortunately, both NUWA(2021) and VDM(2022) didn't release their code or pretrained model. We tried to contact their authors by email or during seminars, however, both of them are unable to release pretrained checkpoints or provide samples given certain prompts. Thus in our original human evaluation we refer to open-sourced and relatively new baselines -- VideoGPT(2021) and **TGANv2 (2020, rather than TGAN in 2017). This also highlights the value of open-sourcing CogVideo for the research community.**
>
> **Part2:** Regarding the resolution difference between models
>
> we choose to preserve the original resolution because resolution is considered an important aspect of video evaluation, and it is another advantage of CogVideo to generate higher-resolution videos than previous works. In the meantime, we ask the evaluators to rate several aspects independently (frame texture, motion realism, semantic relevance), thus the last two aspects can reflect quality other than resolution.
>
> > **Weakness 2:** I suggest authors to compare their model with in other video problems (such as frame conditional video prediction) where there are more baselines to compare to.
>
> Thanks for your suggestions. However, the technical improvements of CogVideo focus on text-video matching: (1) multi-frame-rate training to alleviate misalignment; (2) dual-channel attention to inherit text-image knowledge. Therefore,  tasks such as frame conditional video prediction can not show the improvements of CogVideo. But thanks for your suggestions, we can borrow designs from frame conditional video prediction baselines.
>
> > **Weakness 3:** I suggest authors to perform an analysis on this matter by generating examples which are more detailed and most likely non-existent in the training set.
>
> **More out-of-distribution results:** https://i.imgur.com/M14jOWp.jpg
>
> While our training data only consists of real videos (rather than animations) and a small amount of abstract video (e.g. animation of abstract texture), CogVideo is capable of generating out-of-distribution videos in high quality, including
>
> - Videos not existing in the real world, such as animals acting like human beings.
> - Stylized videos such as watercolor painting and Chinese traditional drawing, while our dataset does not include any stylized videos.
>
> **Analysis:** The out-of-distribution generation capability of CogVideo is two-fold
>
> - Frame-level: generate reasonable frames not existing in the dataset. CogVideo losslessly inherits the OOD frame-level generation capability from CogView2 since it preserves all CogView2's parameters, **showing the superiority of dual-channel attention**.
>
> - Video-level: given OOD frames, CogVideo is able to generate reasonable actions. For example, when given an image of a cat with a red hat playing the guitar, CogVideo can transfer the human hands to cat's paws, and make it pluck the guitar strings.
>
> We must admit that, though, the success rate of extreme out-of-distribution generation is not very high, due to several reasons: 1) Generalization is too hard for extreme out-of-distribution cases, e.g. birds playing guitar. It's difficult to relate birds' wings to human arms. 2) The generalization capability is bounded by the image generation model, and sometimes even the first frame is in poor quality.
>
> > **Weakness 4:** More details for Reproducibility
>
> **Details of the model:** Refer to the section 4, 'Model' part. And all the details (e.g. structures and settings) can be found in our released code.
>
> **Data:** (Besides the description in section 4)  Our data is of similar content and quality to the public dataset WebVid-10M. Every video is paired with a similar caption describing the content, and the length ranges from 1 to 60 sec.  **As WebVid-10M is larger than our dataset and of similar quality, one can reproduce CogVideo with it.** (WebVid-10M was not released at the time we started this project, therefore we used personally collected dataset.)
>
> **Computation:** (Besides the description in section 4) We train the model with a batch size of 416 on 104 A100 GPUs, and the training process for two stages takes ~20 days in total. Without the acceleration including (1) inheriting the text-image knowledge (2) only optimizing the temporal attention channel (3) autoregressive swin attention, it will (1) take ~32 days (60% more) to finish equivalent optimization steps, and (2) converge even slower.
>
> > **Weakness 5**: The quality of the writing can be improved.
>
> Thanks so much for your suggestion! We are adding more introductions to previous works in our final version.

---

> > ### Comment · Reviewer_9Xh6 · 2022-11-20
> > **Thank you for your response.**
> >
> > After reading the authors response I increased my recommendation, given the clarification on OOD generation and reproduciblity.

---

### Official Review · Reviewer_Tzyn · 2022-11-01

**Confidence:** 4
**Correctness:** 3
**Technical Novelty And Significance:** 3
**Empirical Novelty And Significance:** 3
**Recommendation:** 8

**Clarity, Quality, Novelty And Reproducibility:**

The writing is clear and the authors plan to open source the code. Documentation on the creation of the dataset would be helpful, and maybe the dataset itself. Otherwise it will be hard to reproduce. The paper's novelty mainly lies in the scale of the model, the reutilization of the image-text model and the pretraining data. The architectural changes are rather straightforward (good!) but have been introduced in similar form in other works.

**Details Of Ethics Concerns:**

The paper relies on a new video-text dataset and the authors do not describe the data collection process in detail so it is hard to judge. The authors seem aware of ethical issues and point to potential solutions, only some of which seemed to have been implemented when collecting the data.

**Strength And Weaknesses:**

## Strengths
* Solid qualitative results
* Human evaluation
* Good approach aiming at reusing an existing image-to-text model and thereby reutilizing spent compute.
* Simple extension to existing architecture
* Good presentation

## Weaknesses
* Qualitatively the videos show strong artifacts between frames, i.e., unnatural changes between frames. The dynamics are somewhat natural but the consistency in color and form is not very good. This is likely due to the strong prior of the model to draw individual frames (images) because of pretraining. With longer training this might become more natural. Maybe it could even be enforced more strongly. The authors do not address this.
* The model is still clearly limited in its ability to generate natural videos but the authors are not exploring failure cases at all. A chapter or paragraph on limitations would be great.
* No ablation on impact of model and (pretraining) data size.
* Lacking comparisons:
  * No details about how the data was crawled.
  * No comparisons in terms of data (quality/scale) with similar works like Ho et al
  * No comparison to Ho et al on UCF101
  * No human evaluation against Ho et al. which seems comparable in quality
* Seems worse on some benchmarks to more recent work even though these do not have extra pretraining data on text-video.
* Hard to pin-point where improvements come from. When comparing to prior work it is hard to figure out where the main gains come from. Is it the new pretraining dataset, is it the pretrained image-text model, is it the modeling innovations in this paper.

## Questions & Comments
* The finding of FVD on kinetics is interesting! It would be interesting to see the FVD of the reconstruction on the original as well. Could it be that VQ VAE used in this work is not working well?
* From scratch training better than naive fine-tuning? That’s odd.
* Multi-frame rate training evaluation would be better with some quantification. Maybe one could look at perplexities when conditioning on a frame rate by using another ground-truth frame rate and look at the perplexity to see whether the models conditioning is really working.
* Is alpha really important? Why not just taking the sum instead? I strongly suspect it doesn’t matter.
* Time and spatial subsampling is not completely new but was also used for instance in Weissenborn et al 2019. The adaptivity by conditioning is novel though.



**Summary Of The Paper:**

The paper introduces an extension to CogView2, a text to image model, to include a temporal dimension that allows for the generation of short video clips. The authors. The authors extend the architecture by including a temporal attention block in parallel to the spatial attention block pretrained on text->image data. Quantitative results are overall quite good and the human evaluation is very favourable. However, the paper misses some important comparisons and ablations.

**Summary Of The Review:**

The paper is a well written and has solid results. It is definitely a step in the right direction but leaves some open questions and seems to miss some important comparisons. I am nevertheless leaning towards accept.

UPDATE: I read the rebuttal and am mostly happy with it. I think this paper deserves to be presented at ICLR.

---

> ### Author Response · Authors · 2022-11-18
> **Authors' response (3/3)**
>
> > **Question 1:** The finding of FVD on kinetics is interesting! It would be interesting to see the FVD of the reconstruction on the original as well. Could it be that VQ VAE used in this work is not working well?
>
> FVD of the reconstruction on the original:
>
> - UCF-101: 54.22
> - Kinetics600: 19.76
>
> The results show that VQVAE does have a huge influence on FVD performance (The FVD@Kinetics600 is even higher than Video Diffusion Model's!). One possible explanation is that VQVAE is not powerful enough to recover low-level features since its representation is highly compressed, though it works well on recovering human perceptible features.
>
> > **Question 2:** From scratch training better than naive fine-tuning? That’s odd.
>
> We've double-checked the experiment and reported the results truthfully. One possible explanation is that there's a relatively large gap between image generation and original-frame-rate video (not being downsampled) generation, thus **without dual-channel attention** pretrained parameters are dramatically changed during the initial steps. On the contrary, from-scratch training has a (mathematically) better initialization point.
>
> > **Question 3:** Multi-frame rate training evaluation would be better with some quantification. Maybe one could look at perplexities when conditioning on a frame rate by using another ground-truth frame rate and look at the perplexity to see whether the models conditioning is really working.
>
> Thanks for your suggestion and the perplexities are reported as below (evaluated on 1024 samples).
>
> The results suggest that (1) ground-truth samples get the lowest perplexity when conditioned on matched frame rate, and (2) the perplexities become slightly higher when the difference between the conditioning frame rate and ground-truth frame rate becomes larger. Note that we report the perplexities of the first half of the second frame instead of the whole video, because (1) the first frame is independent from frame rate; (2) the changing intensity can be mostly inferred if given the 1st frame and the first half of the 2nd frame, thus the effect of frame rate condition is more prominent on the former tokens.
>
> |                            | Ground-Truth: 1 fps | Ground-Truth: 2 fps | Ground-Truth: 4 fps |
> | :------------------------: | :-----------------: | :-----------------: | :-----------------: |
> | **Model Condition: 1 fps** |       45.878        |       30.008        |       21.050        |
> | **Model Condition: 2 fps** |       46.201        |       29.992        |       20.822        |
> | **Model Condition: 4 fps** |       46.340        |       30.123        |       20.780        |
>
> > **Question 4:** Is alpha really important? Why not just taking the sum instead?
>
> The design of alpha is based on the idea that (1)  initialize CogVideo the same as CogView2 numerically (while simply sum doesn't), (2) explicitly model the difference of impact between temporal and spatial channel thus easing the learning process. According to the 'Attention Analysis' part in Appendix, the value of alpha is learned to be different among layers. Since it's a design detail which only adds a small amount of computation (*hidden_size* parameters for each layer), we didn't spend many computational resources to conduct an ablation study.
>
> > **Question 5:** Time and spatial subsampling is not completely new but was also used for instance in Weissenborn et al 2019. The adaptivity by conditioning is novel though.
>
> Yes, and thanks for acknowledging the novelty of multi-frame-rate training. **It is actually a key design to text-video pretraining**, because the widely collected videos are of various lengths and previous methods failed to align texts and videos. It can be transfered to other models. For example, we're delighted to find that newly released works such as Make-A-Video apply our multi-frame-training in their pretraining.
>
> [1] Yu, Jiahui, et al. "Scaling autoregressive models for content-rich text-to-image generation." *arXiv preprint arXiv:2206.10789* (2022).

---

> ### Author Response · Authors · 2022-11-18
> **Authors' response (2/3)**
>
> > **Weakness 5:** Seems worse on some benchmarks to more recent work even though these do not have extra pretraining data on text-video.
>
> **CogVideo outperforms most models by a large margin in human evaluation and demonstrates state-of-the-art performance on FVD@UCF**, which verifies its effectiveness. But **only on FVD@Kinetics-600** CogVideo underperforms some works, and possible reasons obtained through experiments and analysis are listed below:
>
> 1. Influenced by VQVAE tokenizer. (See section 5.1) CogVideo achieves much better FVD under reconstructed setting.
> 2. Kinetics-600 is of limited domains, while the pretraining corpus covers the general domain, making the impact of pretraining not significant.
> 3. The first 5 frames are given under this evaluation protocol, thus CogVideo's capability of text-to-video generation is not fully demonstrated. Evidence is that CogVideo is the best on FVD@UCF-100, where all frames are generated by models.
>
> > **Weakness 6:** Hard to pin-point where improvements come from. When comparing to prior work it is hard to figure out where the main gains come from. Is it the new pretraining dataset, is it the pretrained image-text model, is it the modeling innovations in this paper.
>
> Thanks for your suggestion! **It is actually a common concern in pretraining, such as DALL-E, Video Diffusion Model,** where datasets, model size and structure are optimized at the same time during final pretraining, and it's too computationally expensive to figure out how much each aspect contributes *at pretraining scale*. **Therefore,  following previous works we try to ablate at an affordable scale to verify the contribution of several aspects**, and the key points are summarized as follows. We have open-sourced CogVideo to enable deeper investigation for future works.
>
> (1) Pretrained image-text model: (section 5.3.1, qualitatively and quantitatively) both finetuning CogView2 and adapting CogView2 with dual-channel attention improves performance, while the latter is better.
>
> (2) The modeling innovations: dual-channel attention (section 5.3.1, qualitatively and quantitatively) and multi-frame-rate training (section 5.3.2, qualitatively).
>
> (3) Dataset size and training time: (section 5.3.1, qualitatively and quantitatively) Training CogVideo with fewer data and steps shows worse performance then original CogVideo.

---

> ### Author Response · Authors · 2022-11-18
> **Authors' response (1/3)**
>
> Thanks for your careful and valuable review! We will address your concerns point by point as follows:
>
> **Summary:** In order to demonstrate our model performance more thoroughly, we additionally conduct human evaluation to compare it with Video Diffusion Model (all 28 text prompts) and NUWA (6 text prompts) (see general response area), where CogVideo achieves the best score over all aspects. We provide analysis on major limitations (weakness 2) and more details (weakness 3). The experiment lacking concern emerges also in previous pretraining works, e.g. DALL-E and Video Diffusion Model, and is usually due to unaffordable computational resources, so we are actively working to conduct affordable ablations to verify the effectiveness of the model innovations instead.
>
> > **Weakness 1:** Qualitatively the videos show strong artifacts between frames, i.e., unnatural changes between frames. With longer training this might become more natural. Maybe it could even be enforced more strongly. The authors do not address this.
>
> As you have suggested, increasing the training time plays an important role in addressing this problem. A simple ablation study is that, we use earlier checkpoints of the Stage 1 model (80,000, 85,000 iterations) to generate samples, and find them to have more severe artifacts. Thanks for your suggestion, and we plan to train for a longer time to remove these artifacts if more computing resources are accessed.
>
> > **Weakness 2:** The model is still clearly limited in its ability to generate natural videos but the authors are not exploring failure cases at all. A chapter or paragraph on limitations would be great.
>
> The major limitations are analyzed below, each attached with possible solutions. And thanks for your suggestion, We have attached it as an additional chapter to the Appendix of our pdf revision.
>
> 1. The quality heavily relies on the first frame, thus CogVideo has limitations similar to image generation model, including unreasonable shape, text-content mismatching, sometimes not robust enough to self-recover. Samples: https://i.imgur.com/uo2FX10.jpg . Increasing data and training for both the based image model and CogVideo may alleviate these problems, and leveraging pretrained text model helps text understanding.
> 2. Slight temporal inconsistency (unnatural changes). Refer to weakness 1.
> 3. Slight blurry induced by VQ-VAE's lossy compression. Our 480\*480-pixel results are decoded from 60\*60 tokens with VQ-VAE, thus may contain slight blur. One possible solution is to further train a super-resolution or deblur model on the pixel level.
>
> > **Weakness 3:** No ablation on impact of model and (pretraining) data size.
>
> **The scaling law of transformer-based text-to-video model has been confirmed by former works**, e.g. Parti[1] , that the performance keeps improving as the model size increases from 350M to 20B (CogVideo is 9B).  **We also set up a small experiment by training a toy version of CogVideo** in sec 5.3.1, which is of the same scale as the original CogVideo but is trained with less data (1 million) and fewer iterations, and qualitatively found that fewer data and training yield worse generation results. But we can't perform many ablations on model scale or data size due to the heavy computational cost of pretraining and the limit of resources (Pretraining CogVideo takes ~20 days on 104 A100 GPUs). **This is actually a common problem encountered by previous works including NUWA, Video Diffusion Model, Make-A-Video, etc.**, which also didn't perform those ablations.
>
>
>
> > **Weakness 4:** Lacking comparisons:
>
> \[Details about how the data was crawled\]: Please refer to "Reproducibility Statement" in our new pdf revision for a detailed description of our dataset. One can refer to WebVid-10M to reproduce CogVideo.
>
> \[Lacking comparison to Video Diffusion Model (Ho et al.)\]: **This is mainly because Ho et al. is not open-sourced,** which also highlights the value of open-sourcing CogVideo for the research community.
>
> - No human evaluation against Ho et al.: **Please check at our general response for a newly added human evaluation with Video Diffusion Model and NUWA, where CogVideo gets better scores on all aspects.**
> - Data scale: We use fewer data (5.4 million text-video pairs) than Ho et al. (10 million captioned videos).
> - Data quality: Ho et al. didn't release their dataset and we didn't find its details in the paper.
> - No comparison to Ho et al. on UCF101: Ho et al. only report FID on UCF-101, which is designed for image evaluation thus is not the focus of video generation. We're unable to evaluate their FVD on UCF-101 by ourselves either because they are close-sourced. Therefore, we choose to compare with Ho et al. on Kinetics-600.

---

### Author Response · Authors · 2022-11-18
**General Response**

We thank all the reviewers for the valuable feedback, and are especially honored for being acknowledged as "one of the first text-to-video generation models with reasonable performance". According to our understanding, there are two main common concerns from reviewers:

1. Comparison with Video Diffusion Model (VDM) and NUWA.

   To address this concern, we strive to demonstrate qualitative comparison and conduct human evaluation with them, *even though they're close-sourced*. The details will be provided later in this general response.

2. The significance of contributions.

   Our contributions are two-fold:

   - Technical contribution.

      (1) propose *multi-frame-rate training* to endow models with a better understanding of text-video relations and abilities to control the intensity of changes during generation. This can be easily transfered to other models, e.g. Make-A-Video apply this technic in their pretraining.

     (2) propose *dual-channel-attention* to efficiently leverage the pretrained text-to-image generative model for a text-to-video generation without hurting its image generation capacity. It not only alleviates video data scarcity but also greatly reduces computational resources (50% less optimized parameters, and reutilizing spent compute) .

   - Community contribution.

     While most mainstream pretrained vision generative models are close-sourced, e.g. Video Diffusion Model, NUWA, Phenaki, CogVideo is an open-sourced work with all the checkpoints and code for training and inference being released (also the first open-sourced pretrained video generation model, to the best of our knowledge). We believe that this will not only set a solid baseline and ease comparison, but also boost the community by providing a chance for everyone to get a hands-on experience with video generative model, and will have a better chance to influence the community if accepted.

## Comparison with Video Diffusion Model and NUWA

In order to demonstrate CogVideo's performance more thoroughly, we additionally conduct human evaluation to compare it with Video Diffusion Model (VDM) and NUWA. Considering both of them didn't release codes or checkpoints (*which is the reason for not including them in our original human evaluation*), we evaluate on all the 28 text-to-video pairs shown on the webpage of [VDM](https://video-diffusion.github.io/) and 6 text-to-video pairs on the webpage of [NUWA](https://github.com/microsoft/NUWA/blob/main/NUWA.md). We invite 21 anonymous evaluators to rate on 3 aspects with score 1-5 (5 indicates the best): overall quality, frame texture and content, motion realism.

**Human evaluation results:**

**CogVideo gets better scores on all three metrics.**

visualization: https://i.imgur.com/hTQJn0d.jpg

- CogVideo vs VDM

  | | Texture & Content | Motion Realism | Overall score |
  | :-: | :-: | :-: | :-: |
  | Video Diffusion Model | 3.30| 3.37 |3.32 |
  | CogVideo (ours) | **4.15**  |  **3.95**  |**4.05** |

- CogVideo vs NUWA

  | | Texture & Content | Motion Realism | Overall score |
  | :-: | :-: | :-: | :-: |
  | NUWA | 3.03  | 2.93 | 2.96 |
  | CogVideo (ours) | **3.76**| **3.91** |**3.89** |

**Qualitative Comparison (all samples in human eval):**

- with NUWA: https://i.imgur.com/d9wmKQh.jpg

- with VDM:

  - (1/4) https://i.imgur.com/GjhG4uN.jpg

  - (2/4) https://imgur.com/u2h8eXa.jpg

  - (3/4) https://i.imgur.com/RHMJwBG.jpg

  - (4/4) https://i.imgur.com/LUB06Og.jpg

 From the samples, we can see that,

1. CogVideo can generate more realistic and detailed objects. E.g. VDM (2/4) "4K illuminated Christmas tree at night during snowstorm. "; VDM (2/4) "Ducks in a pond. "; NUWA "Running on the sea. "; NUWA"A man is folding a piece of yellow paper. ".
2. CogVideo can generate videos with better motion realism and semantic alignment. For example, 1) in VDM (1/4) "pouring coffee into coffee cup", the liquid level keeps raising in CogVideo's sample, while the liquid level sometimes drops in VDM's sample. 2) in VDM (1/4) "sunset at sea", CogVideo shows the whole process of sunset.

## Supplemental Reproducibility Statement

We have paid great exertion to ensure reproducibility, including **open-sourcing code and checkpoints, providing model and training details in the article. As far as we know, CogVideo is the most open text-to-video pretraining work to date.** We choose not to release the datasets in order to ensure copyright issues, according to ICLR Code of Ethics "researchers should therefore respect copyrights, ..., license agreements, and other methods of protecting authors' works". However, **researchers can refer to WebVid-10M for reproducing**, which is a recently released text-video dataset with content and quality similar to our dataset, and larger than ours (10M vs 5.4M). (WebVid-10M was not released at the time we started this project, thus we used personally collected dataset.) **In conclusion, we believe that CogVideo can be easily reproduced.**

---

### Author Response · Authors · 2022-11-19
**Supplemental Ethics Statement**

**=== The complete and detailed ethics statement can be found in our article, page 10, with newly added contents highlighted in blue.**

The primary goal of CogVideo is to advance research on video generation methods. While in the meantime, we are also aware of its possible ethical impact on society: it might be used for malicious purposes. Here we briefly discuss these issues and present possible solutions accordingly.

**Problem 1: Reinforcing social stereotypes.** Visual generation models may inherit biases from their training data and reinforce social stereotypes. For pre-processing, we can fuzzy search keywords related to fairness (such as gender, race, and age) during data collection, and adjust their proportion. For post-processing, we use a Word Replacement solution proposed in Ding et al. (2021) in our developing API.

**Problem 2: Violating privacy.** Researchers have found that some private information contained in the dataset could be extracted from pretrained language models (Carlini et al., 2021). The same problem exists in multi-modal pretrained models. During data collection, we try to filter out data sources with private information, though there inevitably remains a small number of videos of public figures.

**Problem 3: Generating deceptive or harmful content.** We are conscious that pretrained models may generate realistic videos in the near future, which could be used to intentionally misinform subjects or generate sexual and violent videos without adequate guardrails, though there's a certain gap according to our human evaluation. During data collection, we manually filter out sources with inappropriate data including pornographic and violent content, and further filter out toxic texts using stop-word list and NSFW videos using models. When developing API, we set restrictions to prevent users from inputting harmful text descriptions. Additional classifiers will be trained to discriminate the fakes generated by a specific model.

Last but not least, CogVideo is committed to promoting academic research and will never be put into commercial use, and we choose not to release datasets to further ensure copyright issues. Being aware of ethical impacts above, we set a license for CogVideo, which demands the users not to use CogVideo (or derivatives of the model) for any deeds that may violate laws or be harmful to the society.

[1] Carlini, Nicholas, et al. "Extracting training data from large language models." *30th USENIX Security Symposium (USENIX Security 21)*. 2021.

[2] Ding, Ming, et al. "Cogview: Mastering text-to-image generation via transformers." *Advances in Neural Information Processing Systems* 34 (2021): 19822-19835.

---

### Decision · Program_Chairs · 2023-01-20

**Decision:**

Accept: poster

**Justification For Why Not Higher Score:**

The technical contribution is limited, which essentially is a combination of two transformers, one for sequential generation and the other for recursive interpolation.


**Justification For Why Not Lower Score:**

The engineering practices in this paper would be beneficial to the text-to-video community.

**Metareview: Summary, Strengths And Weaknesses:**

This paper was reviewed by five experts in the field. Based on the reviewers' feedback, the decision is to recommend the paper for acceptance to ICLR 2023. All the reviewers acknowledged this work is a solid step towards large-scale text-to-video generation. The reviewers did raise some valuable concerns that should be addressed in the final camera-ready version of the paper, e.g., more details on the dataset collection and implementations should be provided to address the reproduction and ethical concerns. The authors are encouraged to make the necessary changes to the best of their ability. We congratulate the authors on the acceptance of their paper!


**Note From Pc:**

if the above contains the word "oral" or "spotlight" please see: "oral" presentation means -> notable-top-5% and "spotlight" means -> notable-top-25%. As stated in our emails, we are disassociating presentation type from AC recommendations

**Summary Of Ac-Reviewer Meeting:**

N/A